# PNPLA1 has a crucial role in skin barrier function by directing acylceramide biosynthesis

Tetsuya Hirabayashi[1,2,*], Tatsuki Anjo[1,3,*], Arisa Kaneko[1,4], Yuuya Senoo[5], Akitaka Shibata[6], Hiroyuki Takama[6], Kohei Yokoyama[1], Yasumasa Nishito[7], Tomio Ono[7], Choji Taya[7], Kazuaki Muramatsu[3], Kiyoko Fukami[4], Agustí Muñoz-Garcia[8], Alan R. Brash[9], Kazutaka Ikeda[5], Makoto Arita[5], Masashi Akiyama[6] & Makoto Murakami[1,2]

Mutations in patatin-like phospholipase domain-containing 1 (PNPLA1) cause autosomal recessive congenital ichthyosis, but the mechanism involved remains unclear. Here we show that PNPLA1, an enzyme expressed in differentiated keratinocytes, plays a crucial role in the biosynthesis of ω-O-acylceramide, a lipid component essential for skin barrier. Global or keratinocyte-specific *Pnpla1*-deficient neonates die due to epidermal permeability barrier defects with severe transepidermal water loss, decreased intercellular lipid lamellae in the *stratum corneum*, and aberrant keratinocyte differentiation. In *Pnpla1*$^{-/-}$ epidermis, unique linoleate-containing lipids including acylceramides, acylglucosylceramides and (O-acyl)-ω-hydroxy fatty acids are almost absent with reciprocal increases in their putative precursors, indicating that PNPLA1 catalyses the ω-O-esterification with linoleic acid to form acylceramides. Moreover, acylceramide supplementation partially rescues the altered differentiation of *Pnpla1*$^{-/-}$ keratinocytes. Our findings provide valuable insight into the skin barrier formation and ichthyosis development, and may contribute to novel therapeutic strategies for treatment of epidermal barrier defects.

[1] Lipid Metabolism Project, Tokyo Metropolitan Institute of Medical Science, Tokyo 156-8506, Japan. [2] AMED-CREST, Japan Agency for Medical Research and Development, Tokyo 100-0004, Japan. [3] Division of Life Science and Engineering, School of Science and Engineering, Tokyo Denki University, Saitama 350-0394, Japan. [4] Laboratory of Genome and Biosignals, Tokyo University of Pharmacy and Life Sciences, Tokyo 192-0392, Japan. [5] Laboratory for Metabolomics, RIKEN Center for Integrative Medical Sciences, Kanagawa 230-0045, Japan. [6] Department of Dermatology, Nagoya University Graduate School of Medicine, Aichi 466-8550, Japan. [7] Center for Basic Technology Research, Tokyo Metropolitan Institute of Medical Science, Tokyo 156-8506, Japan. [8] Department of Evolution, Ecology, and Organismal Biology, Ohio State University, Ohio 43210, USA. [9] Vanderbilt Institute of Chemical Biology, Vanderbilt University, Nashville, Tennessee 37232-6304, USA. * These authors contributed equally to this work. Correspondence and requests for materials should be addressed to M.M. (email: murakami-mk@igakuken.or.jp).

The skin barrier of terrestrial mammals is essential for prevention of water and electrolyte loss, as well as protection from the penetration of harmful substances and pathogenic microbes[1,2]. Impairment of skin barrier function can cause or aggravate skin disorders, including dry skin, ichthyosis, psoriasis, and atopic dermatitis[3–7]. Although the epidermis is a highly organized stratified epithelium consisting of four distinctive layers, the innermost stratum basale (SB), the stratum spinosum (SS), the stratum granulosum (SG) and the uppermost stratum corneum (SC), its barrier function is provided mainly by specialized structures in the SC and tight junctions in the SG[8,9]. The unique SC components include cross-linked, insoluble proteins of corneocytes forming the cornified envelope (CE) and its associated, external membrane monolayer, called the cornified lipid envelope (CLE), as well as the intercellular lipid lamellae, which are mainly composed of ceramides, cholesterol and free fatty acids (FFAs) and are secreted as lamellar body lipids by differentiated keratinocytes at the SG/SC interface[10–12].

Among the epidermal ceramides with marked molecular heterogeneity (at least 12 classes in humans)[13,14], acylceramide is essential for physical and functional organization of lipids in the SC interstices, and thereby the barrier function of the skin[1,3,11,15]. Impaired biosynthesis or processing of acylceramide causes ichthyosis, characterized by dry, scaly and thickened skin. Acylceramide is an unusual ceramide species whose N-acyl chain is composed of ω-hydroxylated ultra-long chain FAs (ULCFAs) esterified at the ω-position with linoleic acid (LA; C18:2). It has been suggested that the ULCFA portion of acylceramide spans a bilayer while the LA tail inserts into a closely apposed section of bilayer, thus serving as a molecular rivet to link two membranes together in the lipid lamellae[16]. In addition, acylceramides containing ω-O-esterified fatty acids other than LA cannot be converted to covalently protein-bound ω-hydroxyceramide (Cer OS), which forms CLE and functions as a template on the surfaces of corneocytes for the organization of lipid layers in the SC interstices. Indeed, in essential FA deficiency, LA in acylceramide is replaced by oleic acid, which fails to support skin barrier function properly[17]. Until now, several important steps for acyceramide biosynthesis and processing in the epidermis have been identified from studies of autosomal recessive congenital ichthyosis (ARCI) in humans and corresponding mouse disease models with genetic knockouts: the synthesis of ULCFAs by the FA elongase ELOVL4, ω-hydroxylation of ULCFAs by the FA ω-hydroxylase CYP4F22 (or CYP4F39 in mice), and formation of ceramides with ULCFAs by the ceramide synthase CERS3 (refs 18–20). However, the mechanism underlying the formation of acylceramide with LA in the epidermis is still under debate.

The current model for ω-O-esterification of ULCFA-ceramides with LA involves the hydrolysis of triacylglycerol (TG) in lipid droplets to provide LA via a yet unknown lipase, followed by its transfer (via acyl-CoA form) to the ω-hydroxy group of ULCFA in ceramides or glucosylceramides (GlcCer) by a putative ω-O-acyltransferase. Alternatively, LA can be directly transferred from TG to ω-OH Cer and/or ω-OH GlcCer by an LA-specific transacylase[8,21]. Recent studies of patients with neutral lipid storage disease with ichthyosis (NLSDI or Chanarin-Dorfman syndrome) and in mice with Abhd5 deletion suggest that TG accumulation due to loss-of-function of ABHD5 (also known as CGI-58) reduces the availability of LA for acylceramide production[22,23]. ABHD5 is an essential co-factor for stimulation of ATGL (adipose triglyceride lipase, also known as PNPLA2 or iPLA$_2\zeta$), which plays a major role in TG hydrolysis in most tissues[24,25]. ATGL is a member of the patatin-like phospholipase domain-containing protein (PNPLA) or Ca$^{2+}$-independent phospholipase A$_2$ (iPLA$_2$) family, which comprises 9

enzymes in humans acting as lipid hydrolases, acyltransferases or transacylases with diverse substrate specificities including phospholipids and neutral lipids[26]. Interestingly, ichthyosis features and decreased acylceramide levels in the skin have been observed in patients and mice with defective ABHD5 function, but not in those with ATGL mutations or deletion, leading to the proposal that ABHD5 could activate a different lipase that regulates epidermal TG hydrolysis[21,25]. Nonetheless, the molecular entity of ω-O-acyltransferase or transacylase responsible for the linoleoyl ω-O-esterification of ULCFA-ceramides has not yet been identified.

Loss-of-function mutations in PNPLA1, a paralog of ATGL/PNPLA2, have recently been identified in humans or dogs with ARCI (refs 27–29). PNPLA1 fails to hydrolyse TG, however, even in the presence of ABHD5 (ref. 29), raising the question of the role of this functionally orphan enzyme. How lipid metabolism regulated by PNPLA1 contributes to epidermal homoeostasis is a fundamental issue which remains to be addressed. To this end, we herein generated Pnpla1-deficient mice in our ongoing efforts to decipher the biological roles of PLA$_2$-related enzymes by gene targeting[30–33]. We provide evidence that absence of PNPLA1 causes a severe skin permeability barrier defect by perturbing the linoleoyl ω-O-esterification of ceramides to yield acylceramides, along with abnormal differentiation of keratinocytes, thus demonstrating that PNPLA1 is a long-sought enzyme that plays a critical role in acylceramide synthesis in the skin.

## Results

**Expression of PNPLA1 in highly differentiated keratinocytes.** Among adult mouse tissues, Pnpla1 messenger RNA (mRNA) was expressed most abundantly in the skin, followed by the stomach (Supplementary Fig. 1a). Immunohistochemistry of newborn mouse skin revealed localization of PNPLA1 protein in the boundary area between the nucleated SG and the denucleated SC, just above the location of the granular layer marker loricrin, in the epidermis (Supplementary Fig. 1b). In agreement with a previous report[29], PNPLA1 was partially colocalized with filaggrin (a SG marker), but not with keratin 1 and 5 (SS and SB markers, respectively) (Supplementary Fig. 1b). In adult mouse skin, the localization of PNPLA1 in the epidermis was essentially the same as that in newborn skin (Supplementary Fig. 1c). In a monolayer culture of mouse keratinocytes, Ca$^{2+}$ treatment resulted in marked induction of keratinocyte differentiation markers (Krt10 and Lor), as well as Pnpla1 (Supplementary Fig. 1d). Likewise, a marked increase of PNPLA1 expression was observed in human keratinocytes after Ca$^{2+}$-induced differentiation, an event that occurred in parallel with induced expression of the keratinocyte differentiation markers KRT1 and FLG, but not with constitutive expression of the SB marker KRT5 (Supplementary Fig. 1e). These results suggest that PNPLA1 has a specific role in highly differentiated keratinocytes in the uppermost layer of the SG, where lipids required for epidermal barrier function are processed and secreted into the intercellular space to form lipid lamellae and CLE.

**Impaired epidermal permeability barrier in Pnpla1$^{-/-}$ mice.** To gain insight into the function of PNPLA1 in vivo, we generated mice with targeted disruption of the Pnpla1 gene on a C57BL/6 background (Supplementary Fig. 2a,b). The absence of mRNA and protein for PNPLA1 in the skin of Pnpla1$^{-/-}$ mice was confirmed by quantitative PCR (qPCR) (Supplementary Fig. 2c) and immunohistochemistry (Fig. 1a), respectively. Offspring from heterozygote intercrosses were born at the

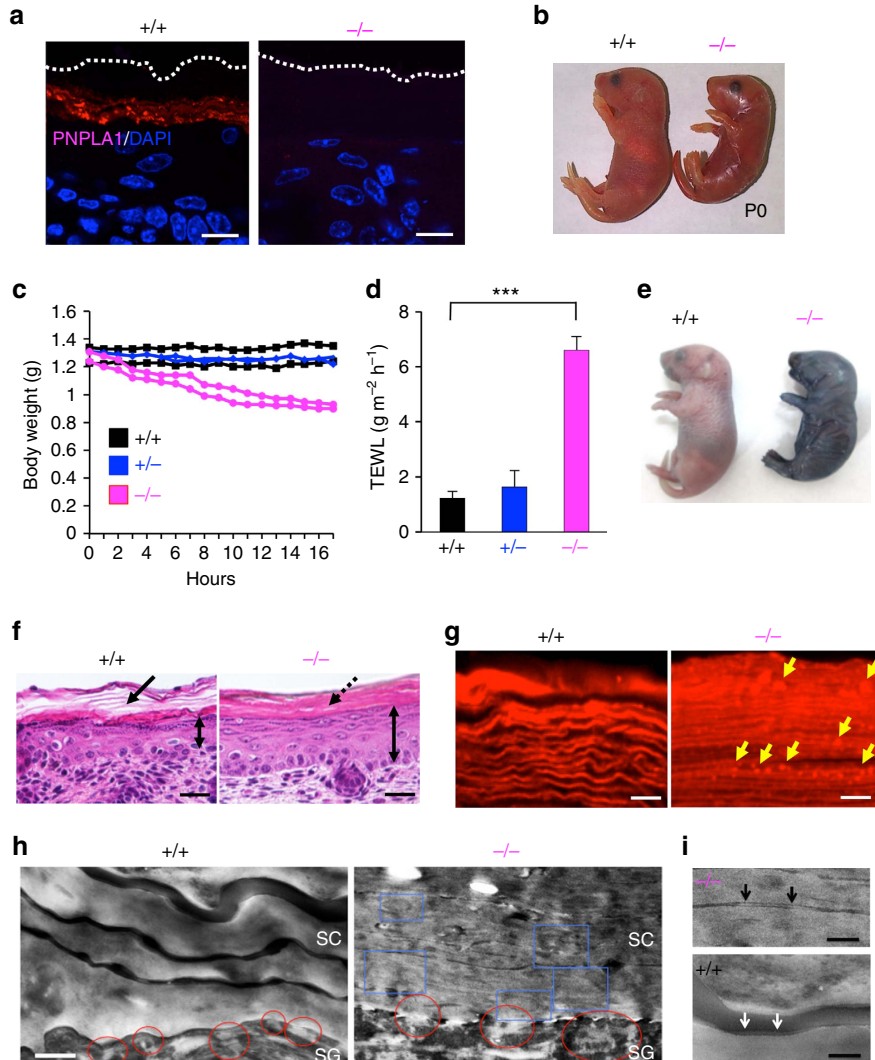

**Figure 1 | Impaired skin barrier function in *Pnpla1*$^{-/-}$ mice. (a)** Immunohistochemical staining for PNPLA1 (red), followed by counterstaining with DAPI (blue), in skin sections from *Pnpla1*$^{+/+}$ and *Pnpla1*$^{-/-}$ newborns. Dashed lines indicate the upper border of the epidermis. **(b)** Gross appearance of *Pnpla1*$^{+/+}$ and *Pnpla1*$^{-/-}$ newborns at P0. **(c)** Monitoring of body weights of *Pnpla1*$^{+/+}$, *Pnpla1*$^{+/-}$ and *Pnpla1*$^{-/-}$ mice (two pups for each) after Caesarean section at E18.5. **(d)** Skin permeability as assessed by TEWL on the dorsal skin surface of *Pnpla1*$^{+/+}$ ($n=19$), *Pnpla1*$^{+/-}$ ($n=21$) and *Pnpla1*$^{-/-}$ mice ($n=21$) (mean ± s.e.m., ***$P<0.001$ in an unpaired, two-tailed Student's *t*-test). **(e)** Toluidine blue exclusion assay using neonatal *Pnpla1*$^{+/+}$ and *Pnpla1*$^{-/-}$ mice. **(f)** Histology of dorsal skin sections from newborn *Pnpla1*$^{+/+}$ and *Pnpla1*$^{-/-}$ mice stained with hematoxylin and eosin. Arrow indicates a basket weave structure in the SC of WT mice. Mutant epidermis was notably thicker (double-headed arrows) and had a more tightly packed SC (dashed arrow) than WT epidermis. **(g)** Staining of SC lipids with Nile red. Continuous linear lipid structures in the SC of *Pnpla1*$^{+/+}$ mice were replaced by a dot-like pattern (yellow arrows) in the SC of *Pnpla1*$^{-/-}$ mice. **(h,i)** Transmission electron microscopy of skins of *Pnpla1*$^{+/+}$ and *Pnpla1*$^{-/-}$ newborn mice. Compared with *Pnpla1*$^{+/+}$ skin, *Pnpla1*$^{-/-}$ skin displayed numerous lipid aggregates in corneocytes (blue boxes), abnormalites in the secreted contents at the SG–SC interface (red circles) **(h)**, and impaired formation of the CLE (arrows) **(i)**. Scale bars; 20 μm **(a,f)**, 5 μm **(g)**, 0.4 μm **(h)** and 0.2 μm **(i)**. Representative **(a–c,e–i)** or complied **(d)** results from at least three experiments are shown.

expected Mendelian ratio (Supplementary Fig. 2d). Although *Pnpla1*$^{+/-}$ mice were healthy and indistinguishable from *Pnpla1*$^{+/+}$ mice, newborn *Pnpla1*$^{-/-}$ pups had shiny and taut skin, often with a necrotic tail tip (Fig. 1b; Supplementary Fig. 2e), and died within 24 h after birth. We hypothesized that the cause of death in these *Pnpla1*-deficient mice might be dehydration, and therefore we assessed their skin permeability barrier function. *Pnpla1*$^{-/-}$ pups delivered by Caesarean section at E18.5 had normal body weight as compared with littermate wild-type (WT) and heterozygous mice at birth, but rapidly lost as much as 20% of their weight within 16 h (Fig. 1c). In accordance with this steep weight loss, trans-epidermal water loss (TEWL) was markedly higher in *Pnpla1*$^{-/-}$ newborns than in *Pnpla1*$^{+/+}$ and

*Pnpla1*$^{+/-}$ newborns (Fig. 1d), indicating a severe defect of the inside-out barrier in the null mice. In the toluidine blue exclusion assay to assess the outside-in permeability barrier, WT littermates excluded dye, whereas *Pnpla1*$^{-/-}$ pups showed robust dye penetration into the skin (Fig. 1e). These phenotypes, which have been commonly observed in mutant mice with disruption of genes associated with ARCI (ref. 1), suggest that PNPLA1 is required for epidermal permeability barrier function.

Histological analysis revealed that control mice had a clear basket weave-like structure segregated by interspaces, indicative of the presence of lipid lamellae (Fig. 1f, left). In contrast, *Pnpla1*$^{-/-}$ mice exhibited a tightly packed structure in the SC, a reduced number of keratohyaline granules in the uppermost SG,

and epidermal hyperplasia (Fig. 1f, right), which is considered to be an adaptive response to barrier disruption. Nile red staining of the $Pnpla1^{+/+}$ epidermis showed wavy lipid multilayers characteristic of SC intercellular lipid lamellae, whereas granular-like lipid aggregates were present within increased number of densely packed lamellar sheets in the $Pnpla1^{-/-}$ epidermis (Fig. 1g), suggesting that keratinocytes are hyperproliferative and defective in the secretion and/or composition of SC lipids in mutant mice. Ultrastructural examination of $Pnpla1^{-/-}$ mice by transmission electron microscopy confirmed the tightly stacked layers of corneocytes with a substantially decreased amount of intercellular lipid lamellae, as evidenced by narrowed SC interstices, in comparison with control mice (Fig. 1h). At the SG–SC interface of $Pnpla1^{+/+}$ epidermis, lipid lamellae were released into the intercellular spaces from lamellar bodies (Fig. 1h). In contrast, the secretion of lipid granule contents was hampered and abnormal vesicular structures, which were thought to represent defective lamellar bodies, were retained within corneocytes in $Pnpla1^{-/-}$ epidermis (Fig. 1h). Moreover, $Pnpla1^{-/-}$ mice exhibited either loss or abnormalities of the CLE (Fig. 1i). These results indicate that PNPLA1 plays a critical role in the proper formation of intercellular lipid lamellae and CLE in the SC, which are important for the permeability barrier function of the skin.

**Aberrant keratinocyte differentiation in $Pnpla1^{-/-}$ mice**. To further address the skin abnormalities in $Pnpla1^{-/-}$ mice, we performed microarray gene profiling using skins of newborn $Pnpla1^{+/+}$ and $Pnpla1^{-/-}$ mice. Heat map visualization of selected genes indicated down-regulation of genes for late keratinocyte differentiation and CE constituents (for example, Lor, Flg, Flg2, and the late cornified envelope genes Lce1a, b) in $Pnpla1^{-/-}$ epidermis (Supplementary Fig. 3a). Most of these are located within the epidermal differentiation complex, a keratinocyte lineage-specific gene locus on mouse chromosome 3. Several up-regulated gene clusters within the epidermal differentiation complex, such as small proline-rich proteins (Sprr1a, Sprr1b and Sprr2b), late cornified envelope proteins (Lce3b and Lce3c), and S100 proteins (S100a8 and S100a9), have been associated with psoriasis[34]. Other up-regulated genes included those involved in keratinocyte proliferation linked to epidermal growth factor (EGF) signalling (for example, Areg, Epgn, Tgfa, Hbegf and Ereg), adhesive structures (for example, Cldn4, Cldn7, Dsc1, Dsc2, Ocln, Tjp1 and Tjp2), lipid metabolism (for example, Fasn, Scd1, Pla2g4d and Pla2g4e) and skin-associated immune responses (for example, Il1b, Il12a, Il13, Il20, Il22, Il23a, Tnf, Ifng and Cxcl1) (Supplementary Fig. 3a). It is likely that the enhanced expression of inflammatory cytokines and chemokines is a secondary effect resulting from impaired barrier function, since similar changes have also been observed in several genetically distinct mouse models with barrier defects[33–38] and patients with skin diseases such as ichthyosis, atopic dermatitis and psoriasis[39,40]. Interestingly, expression levels of genes associated with synthesis and processing of epidermal acylceramide were consistently elevated in $Pnpla1^{-/-}$ mice relative to $Pnpla1^{+/+}$ and $Pnpla1^{+/-}$ mice (Supplementary Fig. 3a,b). These genes included Elovl4, Abhd5, Cers3, Cyp4f39 (a mouse ortholog of human CYP4F22), Ugcg, Abca12 and Gba, mutation or deletion of which has been shown to cause ARCI in humans and neonatal death in mice due to severe skin barrier defects[1,41].

Immunofluorescence staining and qPCR confirmed the diminished expression of terminal differentiation markers, such as filaggrin (Flg) and loricrin (Lor), in $Pnpla1^{-/-}$ skin relative to $Pnpla1^{+/+}$ skin, whereas mRNA and protein expression levels of the basal and early suprabasal keratinocyte differentiation

markers, keratin 5 (Krt5) and 1 (Krt1), were similar between the two genotypes (Fig. 2a,b). In contrast, keratin 6 (Krt6a and Krt6b) was expressed in the lower suprabasal layer in Pnpla1-deficient but not in control skin, reflecting the hyperproliferative state of the mutant epidermis. Abnormal differentiation of keratinocytes has also been observed in several mouse lines with targeted disruption of genes implicated in epidermal ceramide metabolism[22,35]. Therefore, the neonatal lethality of $Pnpla1^{-/-}$ mice due to skin barrier defect is likely dependent upon both altered lipid composition and impaired differentiation of keratinocytes.

Moreover, expression of PPARδ (Ppard) and its potential target genes such as Fabp5 and Sprr1b (ref. 35) was markedly increased in $Pnpla1^{-/-}$ skin relative to WT skin (Fig. 2c; Supplementary Fig. 3a), indicating that PNPLA1 deficiency leads to hyperactivation of PPARδ. Activation of EGF receptors has been shown to control keratinocyte proliferation and differentiation with decreased expression of differentiation-related genes including filaggrin and loricrin[42]. Indeed, heparin-binding EGF-like growth factor (HB-EGF), a potent autocrine growth factor for keratinocytes and putative target gene of PPARδ (ref. 43), was robustly upregulated in $Pnpla1^{-/-}$ epidermis (Fig. 2c; Supplementary Fig. 3a), suggesting that EGF receptor signalling contributes, at least in part, to epidermal hyperplasia and altered keratinocyte differentiation in the mutant mice.

**Defective acylceramide biosynthesis in $Pnpla1^{-/-}$ skin**. To identify the endogenous lipid metabolism regulated by PNPLA1, we performed thin-layer chromatography (TLC) and quantitative liquid chromatography mass spectrometry (LC–MS/MS) using epidermal lipids extracted from neonatal WT and mutant mice. TLC analysis revealed that the bands for acylceramide (esterified omega-hydroxyacyl-sphingosine; EOS), which is a key determinant of skin permeability barrier function[15], and its derivative acylglucosylceramide (GlcEOS) were markedly reduced or almost undetectable in $Pnpla1^{-/-}$ mice relative to WT and heterozygous mice (Fig. 3a). We also noticed that another lipid species, with a TLC motility slightly faster than that of FA, was nearly absent in mutant mice, and LC–MS/MS analysis with collision-induced fragmentation of this lipid extracted from the TLC plate identified it as (O-acyl)-ω-hydroxy FA (OAHFA), particularly (O-linoleoyl)-ω-hydroxy FA (OLHFA) (see below). In contrast, ω-hydroxy FA (ω-OH FA), ω-OH Cer and GlcCer were present in substantially greater amounts in $Pnpla1^{-/-}$ mice than in control $Pnpla1^{+/+}$ and $Pnpla1^{+/-}$ mice (Fig. 3a).

To determine the changes in ceramide molecular species in terms of the length and saturation of their N-acyl chains, lipids extracted from $Pnpla1^{+/+}$ and $Pnpla1^{-/-}$ epidermis were analysed quantitatively by LC–MS and LC–MS/MS. Epidermal ceramide species are grouped into non-hydroxylated ceramides (NS, NDS, NH and NP), α-hydroxylated ceramides (for example, AS, ADS, AH and AP) and acylceramides (for example, EOS, EOH and EOP), where S, P, DS and H stand for sphingosine, phytosphingosine, dihydrosphingosine and 6-hydroxysphingosine, respectively[44,45]. EOS and EOP species with residues of (O-linoleoyl)-ω-hydroxy ULCFAs (C28–C38) were almost entirely lost in the epidermis of $Pnpla1^{-/-}$ mice (Fig. 3b; quantitative data for representative molecular species are depicted in Supplementary Fig. 4a,b). Correspondingly, there was marked accumulation of various molecular species of ω-OH Cer, a putative precursor of EOS, in mutant mice relative to WT mice (Fig. 3c). In addition, in mutant mice, the amounts of OLHFA species with C28–C36 ULCFAs were markedly decreased, with reciprocal increases in corresponding ω-OH ULCFA species (Fig. 3d,e), confirming the results of TLC analysis (Fig. 3a).

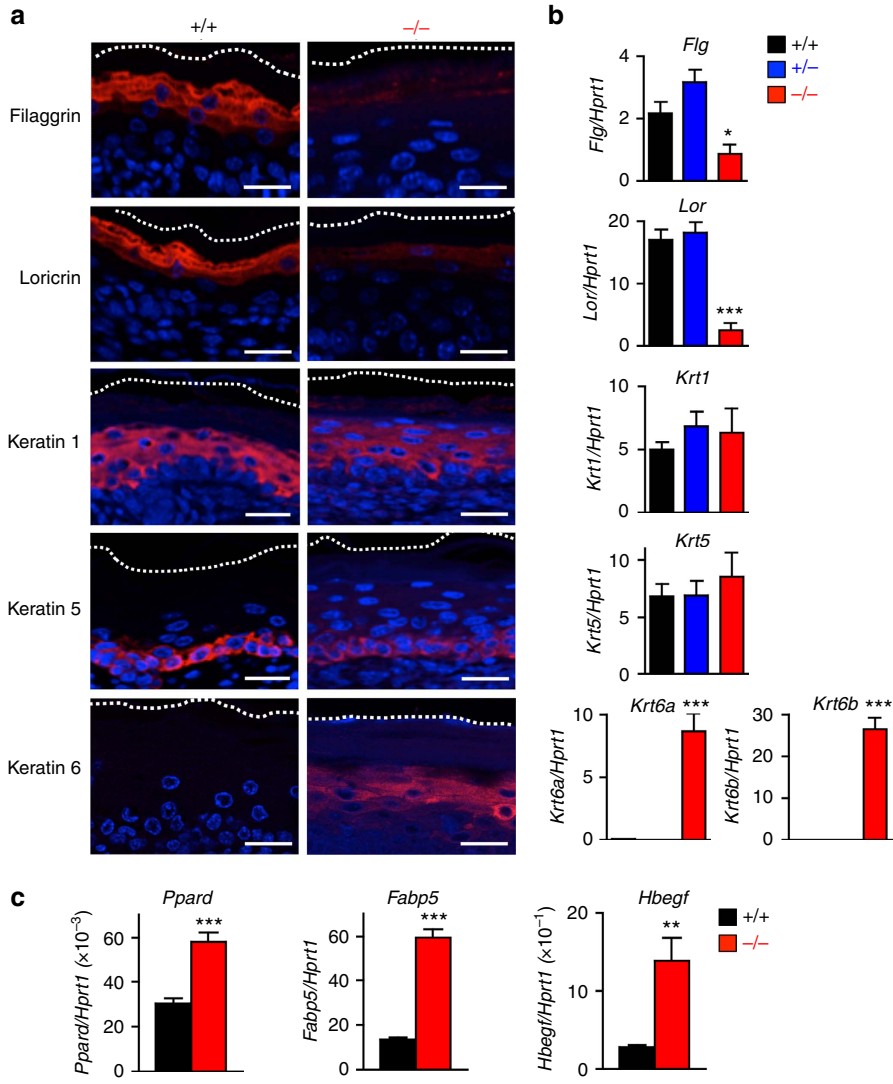

**Figure 2 | Aberrant terminal differentiation of Pnpla1$^{-/-}$ epidermis.** (**a**) Immunohistochemical staining of keratinocyte differentiation markers (red), followed by conterstaining with DAPI (blue), in skin sections from Pnpla1$^{+/+}$ and Pnpla1$^{-/-}$ newborn mice. Scale bars, 20 μm. (**b**) qPCR analysis of keratinocyte differentiation markers in newborn Pnpla1$^{+/+}$, Pnpla1$^{+/-}$ and Pnpla1$^{-/-}$ epidermis (n = 5 animals per group). (**c**) qPCR analysis of PPARδ (Ppard) and its potential target genes in newborn Pnpla1$^{+/+}$ and Pnpla1$^{-/-}$ epidermis (n = 7 animals). In **b,c**, values are mean ± s.e.m.; *P < 0.05, **P < 0.01, and ***P < 0.001 versus Pnpla$^{+/+}$ mice. Representative results from two or three independent experiments are shown.

Moreover, the amount of Cer OS covalently bound to the CE was robustly reduced in mutant mice relative to WT mice (Supplementary Fig. 4c). In contrast to the dramatic reductions of acylceramides and their downstream products, various ceramide molecular species (AS, AP, NS, NH and NP) were modestly increased in Pnpla1$^{-/-}$ mice (Supplementary Fig. 4d–h). Collectively, these data suggest that PNPLA1 is required for linoleoyl ω-O-esterification of the free and/or ceramide-bound forms of ω-OH ULCFA residues. Interestingly, the linoleate residue of several, if not all, EOS and OAHFA species was partially replaced by the palmitate or oleate residue in Pnpla1$^{-/-}$ mice (Fig. 3b,d), indicating that, in the absence of PNPLA1, another putative acyltransferase or transacylase with weak activity and broad substrate specificity may contribute to the synthesis of EOS and OAHFA with non-linoleate fatty acid (that is, palmitate or oleate).

Although the epidermal levels of total ceramides were similar in both genotypes, those of FFAs, cholesterol and TG were substantially higher in Pnpla1$^{-/-}$ mice than in Pnpla1$^{+/+}$ mice (Fig. 3a; Supplementary Fig. 5a). Since a proper ratio of

ceramides, FFAs, and cholesterol is crucial for formation of the SC lipid lamellae, altered proportion of these lipids may lead to their unusual aggregation, as seen in the Pnpla1$^{-/-}$ SC (Fig. 1g). Among the FFAs, the levels of very long chain FAs (VLCFAs; ≥ C22:0), but not those of long chain FAs (LCFAs), were increased in Pnpla1$^{-/-}$ mice (Supplementary Fig. 5b–d). These increases in cholesterol, VLCFAs and several ceramide species resulting from Pnpla1 deficiency accorded with the elevated expression levels of genes related to lipid metabolism such as Hmgcr, Elovl4 and Degs2 (Supplementary Fig. 3a,b), suggesting compensatory adaptation of the Pnpla1$^{-/-}$ epidermis to the impaired acylceramide synthesis and barrier formation. Moreover, our observation that the free LA level was unchanged in Pnpla1$^{-/-}$ mice (Supplementary Fig. 5d) argues against the alternative idea that PLPLA1 acts as a TG lipase that supplies LA for ω-O-esterification of ULCFA. Although the composition of phospholipids was not profoundly affected by Pnpla1 deficiency, some phosphatidylethanolamine (PE) species with polyunsaturated fatty acids, including LA, were present in slightly greater amounts in Pnpla1$^{-/-}$ than in Pnpla1$^{+/+}$ mice

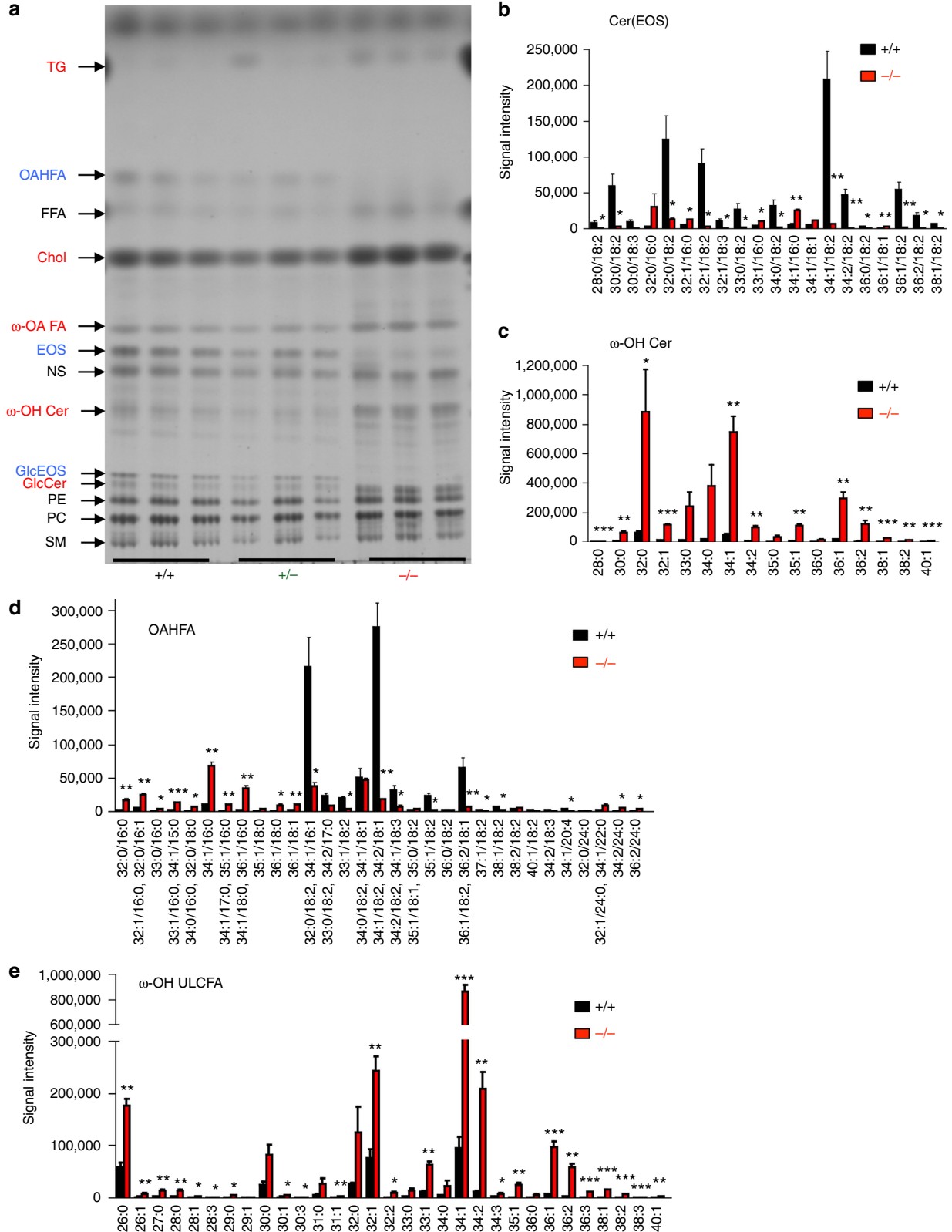

**Figure 3 | Impaired acylceramide formation in *Pnpla1*$^{-/-}$ epidermis. (a)** Representative TLC analysis of lipids extracted from *Pnpla1*$^{+/+}$, *Pnpla1*$^{+/-}$ and *Pnpla1*$^{-/-}$ epidermis. In *Pnpla1*$^{-/-}$ mice, EOS, GlcEOS and OAFHA were almost completely depleted (blue), with reciprocal increases in ω-OH FA, ω-OH Cer and GlcCer (red), relative to *Pnpla1*$^{+/+}$ and *Pnpla1*$^{+/-}$ mice. TG, triglyceride; FFA, free fatty acid; Chol, cholesterol; PE, phosphatidylethanolamine; PC, phosphatidylcholine: SM, sphingomyelin. **(b–e)** LC–MS/MS analysis of epidermal ceramide and related lipid species showing marked reductions in EOS **(b)** and OAHFA **(d)** species with linoleic acid (18:2) and increases in corresponding ω-OH Cer **(c)** and ω-OH ULCFA **(e)** species in *Pnpla1*$^{-/-}$ mice in comparison with *Pnpla1*$^{+/+}$ mice (mean ± s.e.m., $n = 3$ animals; *$P < 0.05$, **$P < 0.01$ and ***$P < 0.001$ versus *Pnpla*$^{+/+}$ mice). In **b,c**, C18-sphingosine-based ceramide (d18:1) species are selected and shown. Results from one or two independent experiments are shown.

(Supplementary Fig. 5e), probably because of the perturbed LA metabolism resulting from impaired formation of OLHFA and acylceramide.

**Keratinocyte-specific *Pnpla1* ablation impairs skin barrier.** To ascertain whether the skin barrier defects observed in global *Pnpla1*[−/−] mice were indeed intrinsic to skin, mice carrying the *loxP*-flanked *Pnpla1* allele (*Pnpla1*[f/f]) were crossed with mice transgenic for *Krt14* promoter-driven Cre recombinase to obtain mice lacking PNPLA1 selectively in epidermal keratinocytes (*Pnpla1*[f/f]*K14-Cre*). Expression of *Pnpla1* in the skin was reduced by ∼80% in *Pnpla1*[f/f] *K14-Cre* mice in comparison with control *Pnpla1*[f/f] mice (Fig. 4a), confirming that Cre-mediated recombination efficiently ablated *Pnpla1* in epidermal keratinocytes.

About half reduction of *Pnpla1* expression was also evident in the stomach, in which the *K14* promoter is active[46], yet it is unlikely that this reduction could influence the skin phenotype since global heterozygous *Pnpla1*[+/−] mice showed no abnormality. Although *Pnpla1*[f/f] *K14-Cre* animals were indistinguishable from control littermates shortly after birth, the mutant mice died within 6 days (Fig. 4b). The death was accompanied by focal desquamation with a markedly elevated TEWL value, whereas the value in other unaffected skin region remained unchanged (Fig. 4c,d). Histologically, a lower zone of the SC layers became densely packed with lipid-poor interspaces in *Pnpla1*[f/f]*K14-Cre* mice at P5 (Fig. 4e), as was seen in global *Pnpla1*[−/−] mice (Fig. 1f). Immunostaining of *Pnpla1*[f/f] *K14-Cre* mice skin demonstrated diminished expression of filaggrin and loricrin in comparison to *Pnpla1*[f/f] mice (Fig. 4f). Furthermore, epidermal

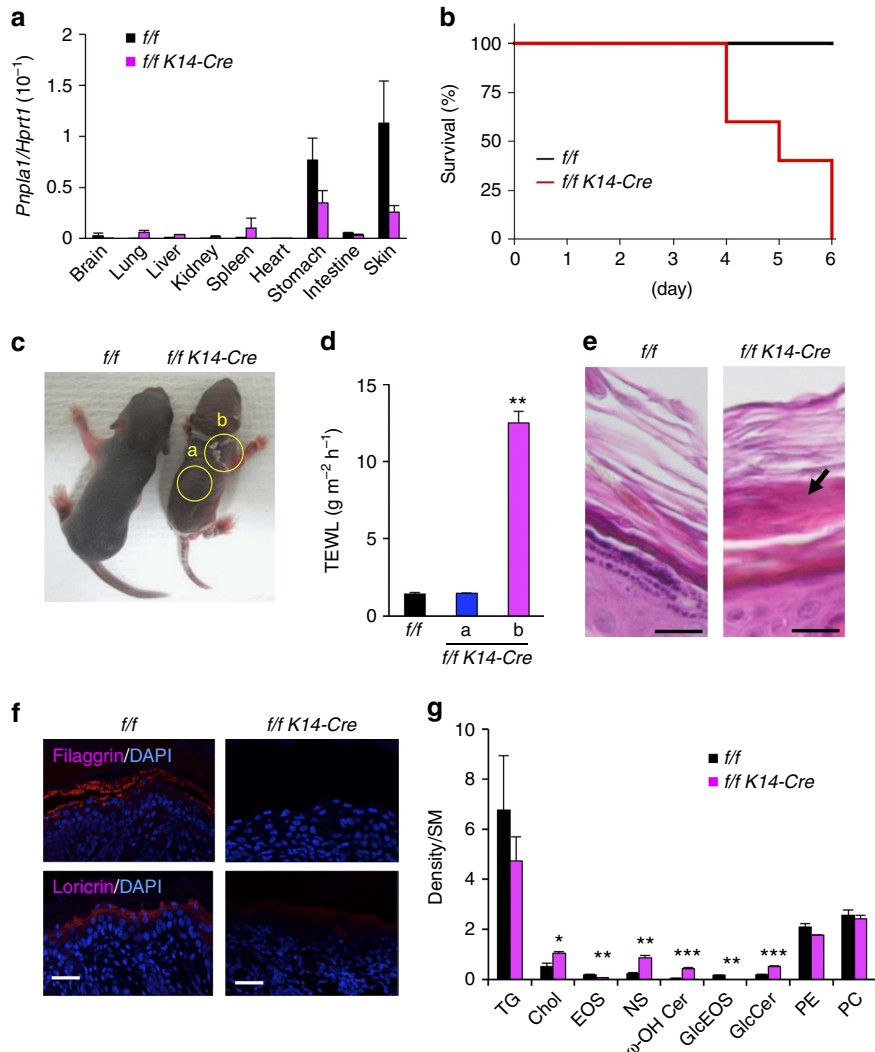

**Figure 4 | Phenotypes of keratinocyte-specific *Pnpla1*-deficient mice.** (**a**) qPCR analysis of *Pnpla1* expression in various tissues of control (*f/f*) (*n* = 2) and *Pnpla1*[f/f] *K14-Cre* (*f/f K14-Cre*) (*n* = 4) mice at P5. (**b**) Postnatal death within 6 days due to epidermal-specific disruption of *Pnpla1* (*n* = 5 per genotype). (**c**) Gross appearance of control (*f/f*) and mutant (*f/f K14-Cre*) mice at P5. Mutant animals showed smaller body size. Yellow circles labelled with **a,b** indicate regions without and with severe desquamation, respectively. (**d**) TEWL of control (*n* = 13) and mutant (*n* = 18) mice at P5. Labels **a,b** are as indicated in **c**. (**e**) Representative images of hematoxylin-eosin staining of skin sections from control and mutant mice at P5. In mutant mice, the lower part of the SC layers became densely packed with poor lipid interspaces (arrow). (**f**) Impaired terminal differentiation of epidermal keratinocytes at P5 in *Pnpla1*[f/f] *K14-Cre* mice. Sections were stained with anti-filaggrin and anti-loricrin antibodies (red) and DAPI (blue). (**g**) Densitometric analysis of TLC separation of epidermal lipids extracted from control and mutant mice at P5. Individual lipid levels were normalized with SM content (*n* = 6 animals). In **a,d,g**, values are mean ± s.e.m.; *P < 0.05, **P < 0.01 and ***P < 0.001 versus control mice. Scale bars in **e,f**, 20 μm. Data are from at least two independent experiments.

levels of EOS and GlcEOS were markedly lower, while those of ω-OH Cer and GlcCer were conversely higher, in $Pnpla1^{f/f}K14\text{-}Cre$ mice (Fig. 4g). Taken together, these results suggest that PNPLA1 is required in a cell-autonomous manner for acylceramide formation and keratinocyte differentiation.

**EOS rescues aberrant differentiation of mutant keratinocytes.** To further investigate the function of PNPLA1 in keratinocyte differentiation, gene expression in primary keratinocytes prepared from $Pnpla1^{-/-}$ and control mice was analysed in culture. Consistent with the *in vivo* data (Fig. 2a–c), expression of the terminal differentiation marker *Flg* was lower in differentiated $Pnpla1^{-/-}$ keratinocytes than in replicate control cells, while that of *Ppard* or *Hbegf* was significantly elevated in differentiated $Pnpla1^{-/-}$ keratinocytes (Fig. 5). Supplementation of the differentiation medium with EOS(C30:0) partially reversed the altered expression of *Flg*, *Ppard* and *Hbegf* in $Pnpla1^{-/-}$ keratinocytes (Fig. 5). These results suggest that the PNPLA1 product EOS or its derivative(s) regulates terminal keratinocyte differentiation partly through modulating PPARδ expression.

**Epidermal lipid composition in $Pnpla1^{-/-}$ and $Abhd5^{-/-}$ mice.** Last, we compared epidermal lipid composition between mouse lines deficient in *Pnpla1* and *Abhd5* (*Cgi58*, a co-factor for a putative TG lipase), both of which appear to converge on the processing of acylceramides[22]. $Abhd5^{-/-}$ epidermis at E18.5 showed partial reductions of OAHFA and EOS, an almost total depletion of GlcEOS, substantial increases of ω-OH Cer and GlcCer, and marked accumulation of TG (Fig. 6a–c). These lipid profiles in $Abhd5^{-/-}$ mice were similar to those in $Pnpla1^{-/-}$ mice, except that TG accumulation was not evident in the latter. Although the increase in TG content in $Pnpla1^{-/-}$ epidermis at P0 may be explained by the induction of lipogenic enzymes (Fig. 3a; Supplementary Fig. 3a), the distinct impact of *Abhd5* and *Pnpla1* ablations on TG levels at E18.5 lends further support to segregation of PNPLA1 from bulk TG hydrolysis in which ABHD5 participates. Nonetheless, the similar reductions of OAHFA, EOS and GlcEOS in both $Abhd5^{-/-}$ and $Pnpla1^{-/-}$ mice support the cooperative roles of ABHD5 and PNPLA1 in the process of ω-*O*-esterification; ABHD5 in assisting the bulk release of FFAs including LA from TG in lipid droplets and PNPLA1 in esterifying part of this LA pool into the free and/or ceramide-bound forms of ω-OH FAs as an acyltransferase or transacylase. Overall, our results provide unequivocal evidence that PNPLA1 is a long-sought enzyme responsible for ω-*O*-esterification in acylceramide biosynthesis leading to proper formation of SC lamellae, keratinocyte differentiation, and thereby skin barrier function (Fig. 6d).

**Discussion**

It is generally known that PLA2 is a group of enzymes that hydrolyse the *sn-2* position of glycerophospholipids to give rise to fatty acids and lysophospholipids. In fact, by hydrolyzing glycerophospholipids, cytosolic PLA2α plays a central role in arachidonic acid metabolism in a wide variety of cells, secreted PLA2s modulate tissue-specific homoeostasis or diseases in given extracellular microenvironments, and PNPLA9 (iPLA2β) and PNPLA8 (iPLA2γ) participate in energy metabolism and neurode-generation[26,47]. However, it has recently become obvious that several members of the PNPLA/iPLA2 family catalyse forms of lipid metabolism other than the typical PLA2 reaction, as exemplified by PNPLA2/ATGL (iPLA2ζ) acting as a major TG lipase in lipolysis and PNPLA3 (iPLA2ε) probably acting as an acyltransferase or transacylase leading to TG accumulation in non-alcoholic fatty liver disease[24,48]. Herein, as part of our ongoing attempts to clarify the biological roles of the PLA2 family using comprehensive gene targeting strategies, we have identified PNPLA1, which represents an ichthyosis-causative gene with unknown function[29], as an enzyme essential for the biosynthesis of acylceramide, a unique lipid component, the presence of which has long been recognized as prerequisite for normal skin barrier function.

Three abundant lipid groups were almost completely absent in $Pnpla1^{-/-}$ epidermis. One of these groups is the acylceramide EOS (and EOP), a key lipid intermediate that is an absolute requirement for formation of the skin barrier and contains saturated, monounsaturated or diunsaturated ULCFA in the *N*-acyl chain and linoleate in the ω-*O*-acyl chain. The second group is GlcEOS, a glucosylated form of EOS, which can be stored in lamellar bodies to be secreted into the intercellular space of the SC and then converted back to EOS by the glucosidase GBA. The third group is linoleate-containing OAHFA (OLHFA), as described below. The corresponding accumulation of putative precursors of these three lipid groups, namely ω-OH Cer, ω-OH GlcCer and ω-OH ULCFA, in PNPLA1-deficient epidermis provides strong evidence that PNPLA1 acts as an ω-*O*-acyltransferase or transacylase required for acylceramide synthesis. In this regard, the accompaning study by Ohno *et al.*[49] has clearly shown that exogenous overexpression of PNPLA1 in cells or PNPLA1-reconstituted proteoliposomes promotes acylceramide formation likely as a transacylase and that *PNPLA1* mutations associated with ARCI inactivate this transacylase activity.

So far, the order and molecular mechnisms by which ULCFAs and specifically LA are hooked onto the ω-OH ULCFAs of (glucosyl)ceramides has not been fully clarified[15,21]. Our new proposed model for epidermal ceramide metabolism is as follows (Fig. 6d): LA is directly tranferred from a linolate-containing TG pool to the ω-OH ULCFA moiety by PNPLA1 as a

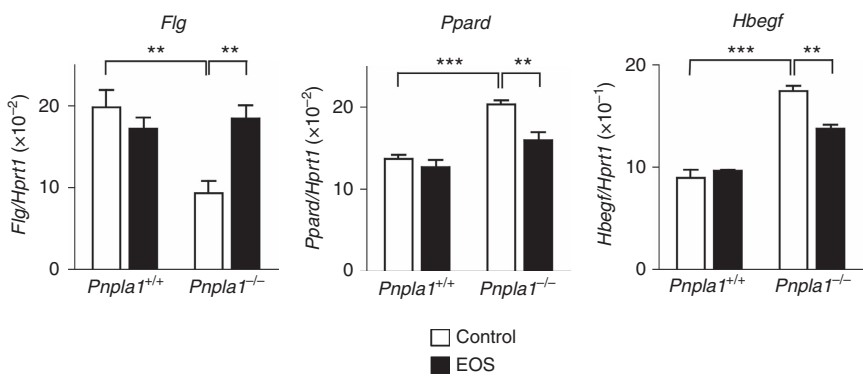

**Figure 5 | Cer EOS partially rescues aberrant differentiation of $Pnpla1^{-/-}$ keratinocytes.** qPCR analysis of gene expression in *Pnpla1*-deficient and control keratinocytes treated with 1.2 mM CaCl₂ for 48 h. EOS was added to the culture medium at 10 μM for the last 24 h. Data are presented as the mean ± s.e.m. ($n = 4$; **$P < 0.01$ and ***$P < 0.001$). Results are representative of two experiments.

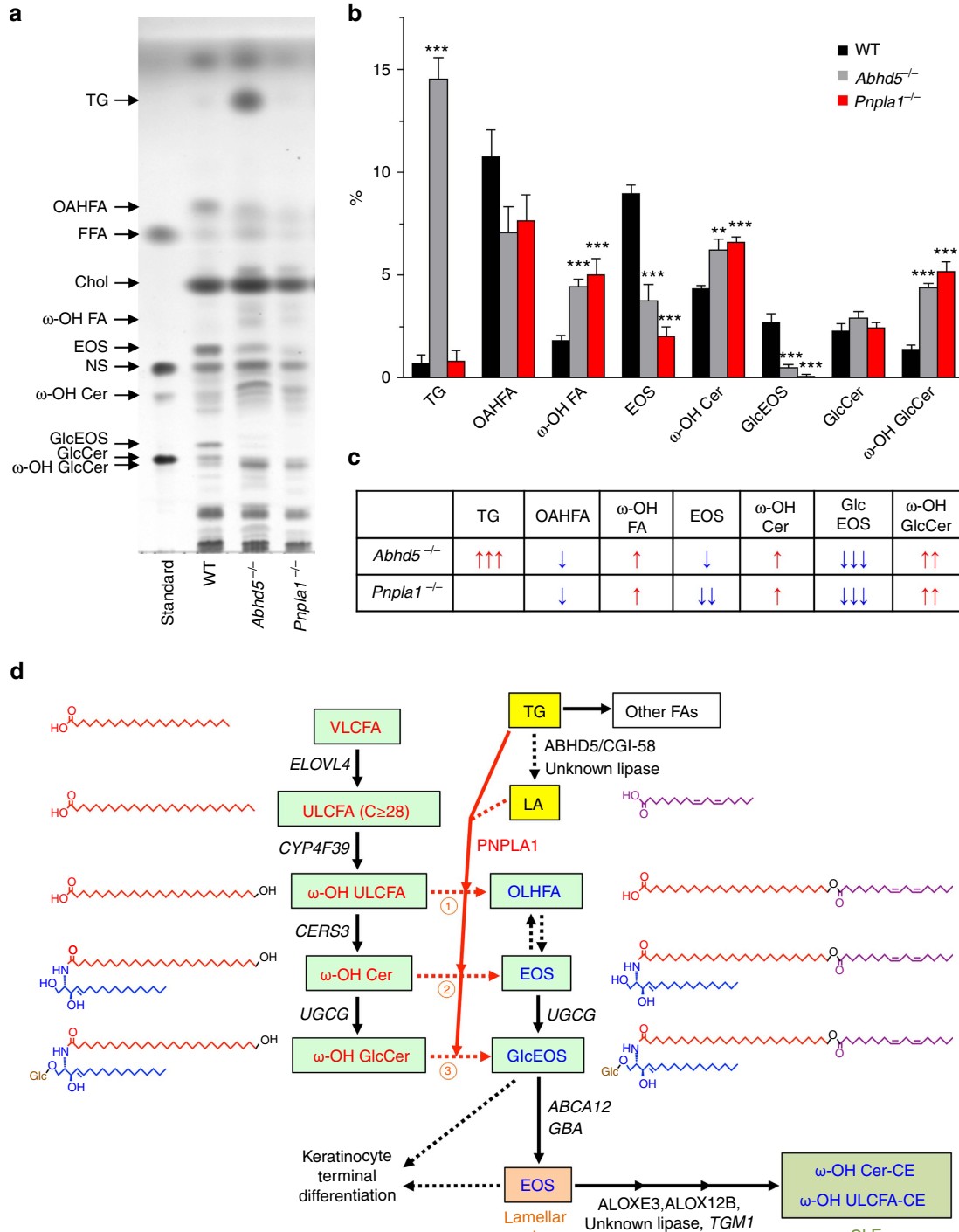

**Figure 6 | Roles of PNPLA1 and ABHD5 in epidermal ceramide metabolism. (a)** Representative TLC analysis of lipids extracted from WT, *Abhd5*$^{-/-}$, and *Pnpla1*$^{-/-}$ epidermis at E18.5. **(b)** Quantification of panel A by densitometric analysis (mean ± s.e.m., $n=6$, 5 and 5 for WT, *Abhd5*$^{-/-}$ and *Pnpla1*$^{-/-}$ mice, respectively; **$P<0.01$ and ***$P<0.001$ versus WT mice). Cumulative results of three independent experiments are shown. **(c)** A summary profile for **a,b**. Up and down arrows represent an increase and a decrease, respectively, in the level of individual lipids in mutant mice relative to WT mice. The number of arrows indicates the relative degree of difference. **(d)** Schematic diagram showing the proposed role of PNPLA1 in epidermal ceramide metabolism in association with keratinocyte differentiation and skin barrier function. Significantly increased and decreased lipid metabolites in *Pnpla1*$^{-/-}$ epidermis are highlighted in red and blue, respectively. Upregulated enzymes are shown in italics. LA derived from TG appears to be esterified at the ω-position of ω-OH ULCFA, ω-OH Cer and/or ω-OH GlcCer (reactions 1, 2 and 3, respectively) by PNPLA1 transacylase. For details, see text.

CoA-independent transacylase to form OAHFA, EOS and/or GlcEOS. Glucosylation of ceramide occurs in the cis-Golgi apparatus through the action of UDP-glucose ceramide glucosyltransferase, UGCG, and then the resulting GlcEOS is incorporated into lamellar bodies and secreted into the intercellular space of the SC. At the SC interstices, the glucosidase GBA deglycosylates GlcEOS to EOS, which form lipid lamellae together with cholesterol and FFA. Two lipoxygenases, ALOX12B and ALOXE3, sequentiallly oxygenate the linoleate moiety in EOS (ref. 50), which then allows a second

as-yet-unidentified lipase to de-esterify acylceramides. The resulting pool of ω-OH Cer can then be covalently linked to the outer suface of the CE, thus foming the CLE.

Although it has been reported that OAHFAs are abundant in the epidermis[51], the function and origin of this unique class of lipids in the epidermis have remained unknown. It is likely that OAHFAs share a biosynthetic reaction with acylceramides, which also contain an N-acyl chain composed of a particular type of OAHFA, namely linoleate-containing OLHFA. There are at least two possible pathways for OAHFA biosynthesis, either directly via ω-O-esterification of ω-OH ULCFA with LA by PNPLA1 (Route 1 in Fig. 6d) or indirectly via synthesis of acyl(glucosyl)ceramides by PNPLA1 and subsequent hydrolysis by a ceramidase (Route 2 and 3 in Fig. 6d). Whether CERS3 could use OAHFA (or its -CoA form) as a substrate for acylceramide synthesis remains to be determined.

The marked alteration of epidermal structure and function along with impaired acylceramide synthesis in $Pnpla1^{-/-}$ newborns, accompanied by down-regulation of CE proteins and up-regulation of EGF ligands, indicate that acylceramide biosynthesis is required for not only the water-impermeable intercellular lipid lamellae in the SC, but also the proper transition from proliferation to terminal differentiation of keratinocytes. The delayed onset of skin phenotypes in $Pnpla1^{f/f}K14$-Cre mice in comparison with global $Pnpla1^{-/-}$ mice may be due to incomplete deletion of cutaneous Pnpla1 expression at birth in the former. The increased expression of PPARδ in $Pnpla1^{-/-}$ epidermis could explain, at least in part, the induction of a panel of lipid metabolism-related genes associated with ARCI. Indeed, PPARδ contributes to up-regulation of ABCA12 and GBA in keratinocytes, and PPARδ deficiency decreases lipid metabolism required for lamellar membrane formation and thereby skin barrier function[52–54]. On the other hand, hyperactivation of PPARδ enhances keratinocyte proliferation through inducing HB-EGF (ref. 43), an event that is recapitulated in $Pnpla1^{-/-}$ keratinocytes.

It is tempting to speculate that the increased extracellular levels of acylceramide or its derivative(s) at the SG/SC border could provide a critical signal for keratinocyte maturation to corneocytes. In our study using cultured $Pnpla1^{-/-}$ keratinocytes, the supplementation with EOS reversed the decreased expression of filaggrin and increased expression of HB-EGF towards normal levels. In support of this observation, application of synthetic pseudo-acylceramide or GlcEOS recovers diminished barrier function in vivo and promotes maturation of cultured keratinocytes by facilitating cornification and CE formation[55,56]. Moreover, markers for keratinocyte proliferation and differentiation are dysregulated in several other knockout mouse lines deficient in the pathway leading to EOS synthesis, processing or transport (for example, $Elovl4^{-/-}$, $Cers3^{-/-}$, $Abhd5^{-/-}$, $Abca12^{-/-}$ and epidermal-specific $Ugcg^{-/-}$)[22,35,41,57,58]. In contrast, keratinocyte differentiation is not profoundly affected in $Alox12b^{-/-}$ mice[59], where protein-bound lipids, but not free ceramides including EOS, are decreased, consistent with the view that the LOX-catalysed oxidation of the linoleate residue in acylceramide is required for subsequent ester hydrolysis and covalent binding of the resultant free ω-OH Cer to the CE[50]. These differences could be explained if differentiated keratinocytes have the ability to sense an extracellular pool of acylceramide or its derivative(s) through a putative receptor, transporter or other way. Nonetheless, the existence of such cross-talk between acylceramide metabolism and transcriptional control of keratinocyte differentiation would be advantageous for the coordinated formation of corneocytes and intercellular lamellar membranes that comprise the SC with competent permeability barrier function, although full

understanding of the underlying mechanism needs further elucidation.

Overall, our analyses of epidermal lipids, morphology and permeability barrier function lend strong support to the contention that PNPLA1 is essential for acylceramide synthesis and skin barrier function. Our genetic approach using knockout mice and the biochemical approach by Ohno et al.[49] complement each other by providing different lines of evidence that prove that PNPLA1 catalyses the ω-O-esterification in acylceramide biosynthesis. While our manuscript was under final review, Grond et al.[60] also reported that acylceramide biosynthesis was impaired in the skin of another $Pnpla1^{-/-}$ mouse strain and in human keratinocytes with PNPLA1 mutation, and that topical application of epidermal lipids from WT mice to $Pnpla1^{-/-}$ skin promoted rebuilding of the CLE. Herein, by means of comprehensive lipidomics, global gene profiling and conditional targeting, we have provided additional insights that the action of PNPLA1 is highly linoleate-selective and keratinocyte-intrinsic. Indeed, ω-O-acyl linoleate in acylceramides and OAHFAs is largely abolished with only partial replacement by other fatty acids in the $Pnpla1^{-/-}$ epidermis, implying that PNPLA1 selectively utilizes linoleic acid for acylceramide biosynthesis and that the loss of this linoleate specificity causes epidermal barrier defect. Although the catalytic mechanism, subcellular localization and functional regulation of PNPLA1 still remain to be elucidated, the findings obtained from these three complementary studies altogether contribute to a better understanding of the skin barrier formation and ichthyosis development, and should be useful in providing novel therapeutic strategies for treatment of patients with skin barrier disorders.

## Methods

**Keratinocyte culture.** Mouse primary keratinocytes were isolated as described previously[61]. Briefly, skins of newborn mice were treated with 5 mg ml$^{-1}$ Dispase (Thermo Fisher) overnight at 4 °C. The epidermis was then mechanically separated from the dermis and incubated with Accutase (Nacalai tesque) for 20 min at room temperature to collect keratinocytes. Human and mouse progenitors for epidermal keratinocytes were purchased from CELLnTEC and have been tested for mycoplasma by the distributor. These cells were cultured in CnT-Prime medium containing 1% (v/v) antibiotic-antimycotic solution (Thermo Fisher). After reaching confluency, the cells were cultured for appropriate periods in CnT-Prime 2D Diff medium supplemented with 1.2 mM CaCl$_2$ to induce keratinocyte differentiation. As required for experiments, Cer EOS (N-(30-Linoleoyloxy-triacontanoyl)-sphingosine; Matreya LLC) was dispersed by sonication for 1 min and then added to the culture. All media were from CELLnTEC.

**Mice.** Pnpla1-deficient mice, containing a 'knockout-first' allele targeted to the Pnpla1 genomic locus named $Pnpla1^{tm1a(KOMP)Wtsi}$, were generated from a conditional targeting vector obtained from the Knockout Mouse Project resource (KOMP-CSD ID:79620) (ref. 62). Briefly, mouse embryonic stem cells derived from C57BL/6N mice (RENKA)[63] containing the correctly targeted Pnpla1 locus were injected into blastocysts and transplanted in pseudopregnant mice to generate chimaera mice. Highly (80–90%) chimeric males were mated with C57BL/6N females, and germ line transmission of the targeted allele was confirmed by PCR. The IRES-LacZ and Neo cassettes were removed by flippase-mediated excision. The male and female heterozygous mice were intercrossed to obtain homozygous null mice, and littermate WT mice were used as controls. Mice with a floxed allele of Pnpla1 were crossed with transgenic mice for K14 promoter-driven Cre recombinase[46] to obtain skin-specific $Pnpla1^{-/-}$ mice ($Pnpla1^{f/f} K14$-Cre).

To generate $Abhd5^{-/-}$ mice, genomic Abhd5 clones were isolated from mouse 129v/Ev genomic library. A 6.7-kb fragment of an Abhd5 clone was subcloned into a targeting vector with exon 1 being replaced by the PGK-Neo cassette. The targeting vector was introduced into 129Sv/Ev embryonic stem cells and a correctly targeted embryonic stem cell line was injected into blastocysts, resulting in the gene-targeted mouse strain. Heterozygotes were backcrossed onto C57BL/6 J background for at least five generations and then intercrossed to obtain homozygous null mice.

Genotyping of offspring was performed by PCR of tail-snip DNA using genotyping primers (Supplementary Table 1). Animals were fed ad libitum (CE2, Clea Japan), had free access to water, and were kept on a 12:12-h light:dark cycle in single cages. All experimental procedures involving animals in this study were approved by the Institutional Animal Care and Use Committees of Tokyo Metropolitan Institute of Medical Science and Nagoya University and were

conducted in accordance with the Japanese Guide for the Care and Use of Laboratory Animals.

**Skin permeability assays.** Toluidine blue staining of newborn mice was described previously[22]. In brief, newborns were anesthetized, dehydrated in methanol, washed in PBS and stained for 30 min in 0.1% (w/v) toluidine blue/PBS. After washing in PBS, the pups were photographed. TEWL was measured using a Tewameter TM300 with a 2-mm-diameter adaptor (Courage-Khazaka Electronics, Germany). Measurements were performed after calibration of the device at room temperature with minimized influence of air turbulence, and the results were recorded when the TEWL values were stabilized 30–45 s after probe placement.

**Histology and immunohistochemistry.** Skins were frozen and embedded in OCT compound, or fixed overnight with buffered 4% (w/v) paraformaldehyde at 4 °C and embedded in paraffin. Skin sections (5 μm) were stained with hematoxylin and eosin or processed for immunoreactions. For immunofluorescent staining, the deparaffinized sections were blocked with PBS containing 10% (w/v) Block Ace (Sumitomo Dainippon Pharma), incubated with primary antibodies overnight at 4 °C and with secondary Alexa Fluor antibodies at room temperature for 1 h. Rabbit polyclonal antibody against PNPLA1 was raised against a synthetic peptide GPPVEDLGPERPTATGSP as an immunogen and used at 1:2,000 dilution. Other primary antibodies used at 1:1,000 dilution were rabbit antibodies against filaggrin, keratin 1, keratin 5, keratin 6 and loricrin (PRB-417P, PRB-149P, PRB-160P, PRB-169P and PRB-145P; Covance). Secondary antibodies used at 1:1,000 dilution were Alexa Fluor 546 goat anti-rabbit IgG and Alexa Fluor 633 goat anti-rabbit IgG (Thermo Fisher). For double immunostaining, rabbit polyclonal antibodies were pre-labelled with Alexa Fluor 488 or Alexa Fluor 555 using the Zenon Rabbit IgG Labeling Kits (Thermo Fisher). After immunoreactions and counterstaining with DAPI (Vector Laboratories), the slides were visualized using an LSM710 confocal microscope (Zeiss). For immunostaining of adult mouse skin, the sections were incubated with anti-PNPLA1 antibody, treated with biotinylated anti-rabbit antibody (BA-1000, Vector Laboratories), and then processed with an avidin–biotin–peroxidase system (Vectastain Elite ABC kit, Vector Laboratories) and diamino-benzidine, followed by counterstaining with hematoxylin. Lipids were visualized on cryosections by staining with Nile Red (5 mg ml$^{-1}$ in 75% (v/v) glycerol, Wako).

**Electron microscopy.** Neonatal mouse skin samples were fixed in 5% (w/v) glutaraldehyde solution, post-fixed in 0.5% (w/v) ruthenium tetroxide (RuO$_4$), dehydrated and embedded in Epon812 (TAAB Laboratories). All the samples were ultra-thin sectioned at a thickness of 70 nm, and stained with uranyl acetate and lead citrate. Photographs were taken using a JEM1400 transmission electron microscope (JEOL Ltd.).

**Quantitative PCR.** Total RNA was extracted with TRIzol (Thermo Fisher) and was reverse-transcribed into cDNA using ReverTra Ace qPCR RT Master Mix (Toyobo) in accordance with the manufacturer's instructions. qPCR reactions were performed on a LightCycler480-II (Roche) using THUNDERBIRD Probe qPCR Mix (Toyobo). The sequences of primers designed to be compatible with the Roche Universal Probe Library (UPL) are provided in Supplementary Tables 2 and 3. The hydrolysis probe used in the assay was labelled with a fluorescein-based reporter dye (FAM) and a non-fluorescent quencher. Cycling conditions were the following: 95 °C for 15 min (one cycle), 95 °C for 15 s and 60 °C for 1 min (40 cycles). A total of 1–2 μl of cDNA per sample was used for the quantification of endogenous mRNA levels. Expression levels were normalized to *Hprt1* or *RPL13A*.

**Microarray.** Total RNA extracted from P0 newborns was purified using a RNeasy Mini Kit (QIAGEN). The quality of RNA was assessed with a 2100 Bioanalyzer (Agilent Technologies). Fluorescently labelled antisense RNA (cRNA targets) were synthesized with a Low Input QuickAmp Labeling Kit according to the manufacturer's protocol (Agilent Technologies). Samples were hybridized to the Mouse Gene Expression 4x44K v2 Microarray (G4846A, Agilent Technologies), washed, and then scanned using a SureScan Microarray Scanner (Agilent Technologies). Microarray data were analysed with Feature Extraction software (Agilent Technologies) and then imported into GeneSpring GX software (Agilent Technologies). Signal intensities were normalized by global normalization.

**Lipid analysis.** After subcutaneous tissue was removed by scraping on ice, skin pieces were incubated in phosphate-buffered saline at 60 °C for 1 min or in phosphate-buffered saline containing 1.5 mg ml$^{-1}$ dispase (Invitrogen) at 4 °C overnight, and the epidermis was peeled from the dermis. The isolated epidermis was vortex-homogenized with steel beads in 1 ml of methanol using beads crusher μT-01 (TITEC). Free lipids were extracted by a modified Folch method[64,65] or with a series of mixtures of chloroform/methanol 2:1, 1:1 and 1:2 (v/v). Combined organic phases were washed once with 0.5 ml of 0.88% KCl and twice with distilled water and then dried with a nitrogen stream. Covalently bound lipids were extracted by incubating the remaining tissue in 2 ml of 1 M NaOH in 90% (v/v) methanol at 60 °C for 2 h. After adjusting the pH to 6 with 3 M HCl, lipids were extracted twice with chloroform. The organic layer was washed twice with distilled

water. The amount of dry lipids was calculated by subtracting the weight of the empty vials.

**TLC.** Epidermal lipids corresponding to 5 mg dry weight were separated by TLC (Silica gel 60, Merck) with the following solvent sequence: 1) chloroform/metha-nol/water (40:10:1) to 2 cm; 2) chloroform/methanol/water (40:10:1) to 5 cm: 3) chloroform/methanol/acetic acid (47:2:0.5) to 8.5 cm; 4) n-hexane/diethyl ether/acetic acid (65:35:1) to the top of the plate. Lipids were visualized after treatment with 5% (w/v) CuSO$_4$ in 15% (v/v) H$_3$PO$_4$ and heating to 180 °C for 10 min. Lipid classifications were performed by comparison with authentic lipid standards or by LC–MS/MS analysis. The intensities of bands were quantified by densitometry using LAS-4000 imaging system (Fuji Film) and JustTLC software (Version 4.0.3, Sweday).

**MS analysis.** LC–MS/MS and LC–MS were used for identification and quantification of epidermal lipids. Non-targeted lipidomics analysis[65,66] were performed with minor modifications. Briefly, dried total lipid extracts were re-dissolved in 50 μl of chloroform:methanol (2:1, v/v) and 2 μl of samples were separated by an ACQUITY UPLC BEH C18 column (50 × 2.1 mm i.d., particle size 1.7 μm, Waters) at a flow rate of 300 μl min$^{-1}$ at 45 °C using an ACQUITY UPLC system (Waters) equipped with a binary pump and automatic sample injector. Solvent A consisted of acetonitrile/methanol/water (20:20:60, v/v/v) and solvent B was isopropanol, both containing 5 mM ammonium acetate. The solvent composition started at 100% A for the first 1 min and was changed linearly to 64% B at 7.5 min, where it was held for 4.5 min. The gradient was increased linearly to 82.5% B at 12.5 min, followed by 85% B at 19 min and 95% B at 20 min before re-equilibrating the column with 100% A for 5 min. Qualitative and quantitative analysis of lipids was performed by MS and data-dependent MS/MS acquisition with a scan range of m/z 70–1250 using a Triple TOF 5600$^+$ System (AB SCIEX) in the negative and positive ion mode. Raw data files from the TOF-MS were converted to MGF files using the program AB SCIEX MS converter for subsequent quantitative analysis with 2DICAL (Mitsui Knowledge Industry). Identification of molecular species was accomplished by comparison with retention times and MS/MS spectra with commercially available standards or reference samples.

Ceramides, VLCFAs, and cholesterol were quantified by LC–MS using an Agilent 1100 Series LC/MSD SL system equipped with a multi-ion source, ChemStation software, an autosampler and an L-column ODS (150 × 2.1 mm i.d.; Chemicals Evaluation and Research Institute)[20,67]. Briefly, the lipid extracts supplemented with the internal standard C17:0 ceramide were dried under a nitrogen stream and then were dissolved in chloroform/methanol/2-propanol (10:45:45, v/v/v). Lipid sample of 20 μl was injected and separated by reversed-phase chromatography at a flow rate of 0.2 ml min$^{-1}$ using a binary gradient solvent system: Solvent C consisted of methanol/water (1:1, v/v) and solvent D was 2-propanol, both containing 5 mM acetic acid and 10 mM ammonium acetate. The column temperature was maintained at 40 °C and the mobile phases were consecutively programmed as follows: 0–1 min, 20% D; 1–2 min, gradient to 60% D; 2–30 min, gradient to 100% D; 30–35 min, 100% D; 35–45 min, 20% D. MS parameters were as follows: negative ion mode, flow of heated dry nitrogen gas 4.0 l min$^{-1}$, nebulizer gas pressure 60 psi, heater temperature of nitrogen gas 350 °C, vaporizer temperature 200 °C, capillary voltage 4,000 V, charging voltage 2,000 V and fragmenter voltage 200 V. Each ceramide species was detected by selected ion monitoring as m/z [M + CH$_3$COO]$^-$.

Analysis of phospholipids and MCFAs was performed using a 4000 QTRAP quadrupole-linear ion trap hybrid MS (AB Sciex) with liquid chromatography (LC-20AP; Shimadzu)[33]. The internal standard mixture added to each sample (equivalent to 2 mg dry weight skin) contained 400 pmol of PE 28:0, 100 pmol of phosphatidylcholine (PC) 28:0 and 100 pmol of lysoPC 17:0. Sample (10 μl) was injected by an autosampler and separated using a Develosil C30-UG column (150 × 1.0 mm i.d., particle size 3 μm, Nomura Chemical) by a step gradient at a flow rate of 80 μl min$^{-1}$ at 50 °C. Solvent E consisted of acetonitrile/methanol/water (1:1:1, v/v/v) and solvent F was 2-propanol, both containing 5 μM phosphoric acid and 1 mM ammonium formate. Lipid peaks were identified according to retention times and multiple reaction monitoring transitions, and quantified by comparison with standard curves using the peak area ratio method.

Covalently bound ceramide extracts were separated by straight-phase HPLC and analysed using Thermo Finnigan DSQ instrument with an APCI ion source operated in positive mode. Sample of 5 μl was injected into a TLC Advantage Silica column (250 × 4.6 mm i.d., 5 μm particle size, 150 Å pore size; Thomson Instrument Company) and separated with hexane/isopropanol/acetic acid (90:10:0.1, vol/vol/vol) at a flow rate of 1 ml min$^{-1}$. Settings in the DSQ were as follows: capillary temperature 275 °C, ion transfer voltage 2,000 V, vaporizer temperature 450 °C, gas 1 set at 50, gas 2 set at 5, electron energy 70 eV and full scan m/z range 400–1,400. The spectra were obtained in full scan mode.

**Statistical analysis.** Sample sizes were chosen based on previous experience in our laboratory. The experiments were performed and analysed in non-randomized and non-blinded fashion. No data were excluded from the analysis. Significance was determined by unpaired two-tailed Student's *t*-test. Variance was similar

between the groups that were statistically compared. A $P$ value of $<0.05$ was considered statistically significant. All the data were presented as mean ± s.e.m.

**Data availability.** The data that support the findings of this study are available from the corresponding author on request. The microarray data can be accessed at the GEO repository under the accession number GSE87682.

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

## Acknowledgements

We thank Drs A. Naoe, H. Tsujimura and J. Ishikawa (Kao Corporation, Japan) for lipid analysis, Drs I. Kawashima (Tokyo Metropolitan Institute of Medical Science, Japan) and A. Yamashita (Teikyo University, Japan) for technical advice, Drs S. Arata and A. Ohazama (Showa University, Japan) for providing *K14-Cre* transgenic mice, and the other members of the laboratory for discussions and comments. This work was supported by JSPS KAKENHI Grant Numbers JP15H05905, JP15K14957, JP16H02613 (to M.M.) and JP15K15094 (to T.H.), AMED-CREST from Japan Agency for Medical Research and Development (to T.H. and M.M.), Kao foundation (to M.M.) and NIH grant AR51968 (to A.R.B.).

## Author contributions

T.H. and M.M. designed experiments. T.H., T.A., A.K., Y.S., A.S., H.T., K.Y., Y.N. and A.M.-G., performed the experiments. T.H., T.A., Y.S., K.M., K.F., A.R.B., K.I. Makoto Arita and Masashi Akiyama analysed the data. T.O. and C.T. designed and constructed mutant mice. T.H. and M.M. wrote the manuscript with input from all the other authors.

## Additional information

**Competing financial interests:** The authors declare no competing financial interests.

