## [Peer Review File · Nature Communications]

Reviewers' comments:

Reviewer #1 (expert in lipid metabolism)

Remarks to the Author:

In the manuscript by Hirabayashi et al., the authors investigated the possible role of PNPLA1 in acylceramide generation and attempted to uncover its role in autosomal recessive congenital ichthyosis. Using a genetic knockout model of PNPLA1, the authors present data showing that PNPLA1 KO mice have skin defects and that these defects might be associated with the loss of ultra long chain hydroxyl omega-acylceramides. While some of the data presented are novel and interesting, there are significant deficits in the manuscript to allow concluding that PNPLA1 is the enzyme generating OLHFA, EOS, and GlcEOS. In addition, the manuscript is lacking mechanistic insights into how PNPLA1 products can lead to proper skin keratinocyte differentiation. Below are point-by-point comments:

Major points:

The data presented in the manuscript are indicative of the regulation of a function in the proposed pathway (Figure 6d). However, the authors should carry out in vitro activity assays to show that indeed PNPLA1 can utilize omega-OH FA, omega-OH Cer, and omega-OH GlcCer to generate OLHFA, EOS, and GlcEOS, respectively.

Mechanistically, how do the proposed PNPLA2 generated products (OLHFA, EOS, and GlcEOS) regulate keratinocyte terminal differentiation? Can these products compensate for each other for the downstream signaling?

What is the mechanism controlling the observed increase in mRNA levels of lipid metabolism related genes associated with ARCI?

Other points:

In Figure 1 localization of PNPLA1 in newborn skin was shown, does the localization change in adult skin tissue?

Fig 2F- Why does the PNPLA1 KO have thicker epidermis?

Fig 2G- Comment on increased number of lamellar sheets in PNPLA1 KO mutant?

Fig 3A- The immune response related genes are not relevant in this figure.

Fig S3 -Viral transduction of PNPLA1 in air-lift cultures of PNPLA1 KO cells should be performed to rescue the effects on Lucifer yellow and H&E staining.

Fig 4A- What is the reason for elevated cholesterol and TG in PNPLA1 KO?

Page 11, 2nd paragraph talking about cholesterol and FFA should be fig S5a not 5a.

Fig 5a- Stomach mRNA levels should be included to show the specificity of the gene targeting since stomach has the second highest expression in WT (Fig1a).

In Figure 4a there is clear increase in TG levels in PNPLA1 KO, but in figure 6a there is a minute increase in TG in PNPLA1 KO. These two figures are not consistent.

Reviewer #2 (expert in skin barrier)
Remarks to the Author:

The manuscript by Hirabayashi et al., titled "PNPLA1 has a crucial role in skin barrier function by directing acylceramide biosynthesis" has been reviewed.

The authors demonstrate PLPLA1 deficiency leads to neonatal lethality in mice, strongly correlated with skin barrier defects and change in lipid composition and ultrastructural evidence for altered lipid lamellar membranes. Lipid composition profile revealed decrease acylceramides (AcylCer), acylglucosylceramides (AcylGlcCer), and O-acyl-omega-hydroxy fatty acids, as well as increase in presumed precursor lipid species (omega-hydroxy GlcCer) and omega-hydroxy fatty acids. The authors conclude that PNPLA1 catalyzes the omega-O-esterification step (with linoleic acid) to form critical AcylCer in the epidermis.

The results and presentation are both high quality, and the work is original and highly significant. Authors generated a PNPLA1 knockout mouse that showed neonatal lethality, with severe skin defect, altered omega-O-esterification (reduced O-acyl-ceramides), and abnormal epidermal differentiation. Combined data demonstrate the requirement for PNPLA1 in generation of acylCer in epidermis, one of the last key steps in the formation of these unique and critical lipids. Statistics and references cited appear appropriate, and conclusions justified by data presented.

General Comments:

Manuscript appears to have dual focus: 1) effect of decreased PNPL1 on epidermal function and lipid composition, revealing key enzymatic activity; and 2) characterization of the PNPL1 knockout. Perhaps better if authors could focus manuscript on item #1; this would reduce need for much of the differentiation-related information that appears to distract from key point of manuscript.

Altered epidermal differentiation can itself lead to altered skin barrier function. The authors should comment that the neonatal lethality is likely dependent upon both factors (altered lipid content and abnormal differentiation) as potentially independent events.

Authors did not directly address mechanism by which lack of PLPLA1 alters the differentiation pattern and expression of differentiation-related genes; as such, the question of possible additional activities of PLPLA1, or action(s) of the products of its now apparent enzymatic activity in epidermis, or that of an accumulating precursor, contributing to the skin abnormalities remain unresolved. Inclusion of appropriate comments on this issue appear necessary.

Results with ABCA12 (-/-) appears to add a layer of complexity to this work that is not critical for the above-noted focus (on PNPL1), especially for the more general audience; suggest removal of these data and discussion to focus manuscript.

Minor comments:

Figure 1: these data are largely represented in Figure 2a (+/+); i.e., showing localization. As such, Fig 1 could be moved to supplemental data.

Figure 3a: Move to supplemental data; Move Supplemental Fig 2b to Fig 3b (add on).

Supplementary Figure 3: Unclear why reconstituted (-/-) shows no evidence of hyperplasia. Suggest

remove this figure.

Figure 4f: Remove or move to supplementary data, as does not appear to add significantly.

Responses to Reviewers

Responses to reviewers:

Reviewer #1 (expert in lipid metabolism)

Remarks to the Author:

In the manuscript by Hirabayashi et al., the authors investigated the possible role of PNPLA1 in acylceramide generation and attempted to uncover its role in autosomal recessive congenital ichthyosis. Using a genetic knockout model of PNPLA1, the authors present data showing that PNPLA1 KO mice have skin defects and that these defects might be associated with the loss of ultra long chain hydroxyl omega-acylceramides. While some of the data presented are novel and interesting, there are significant deficits in the manuscript to allow concluding that PNPLA1 is the enzyme generating OLFHA, EOS, and GlcEOS. In addition, the manuscript is lacking mechanistic insights into how PNPLA1 products can lead to proper skin keratinocyte differentiation. Below are point-by-point comments:

[Answer] Thank you very much for your critical comments and useful suggestions, which were very helpful to improve and strengthen our study. According to your comments, we have amended our manuscript as follows.

Major points:

1) The data presented in the manuscript are indicative of the regulation of a function in the proposed pathway (Figure 6d). However, the authors should carry out *in vitro* activity assays to show that indeed PNPLA1 can utilize omega-OH FA, omega-OH Cer, and omega-OH GlcCer to generate OLHFA, EOS, and GlcEOS, respectively.

[Answer] We agree that the *in vitro* enzymatic assay is important. Actually, we performed a lot of experiments to measure the enzymatic activity of recombinant PNPLA1, but such trials were unfortunately unsuccessful due to the difficulty to solubilize PNPLA1 protein from the membrane fraction. Meanwhile, the back-to-back study by Kihara's group has clearly demonstrated that PNPLA1 acts as a transacylase that transfers linoleate to omega-OH-ceramide to give rise to acylceramide in an *in vitro* enzyme assay using a proteoliposome system.

Since the editor says "though we share the view of Referee #1 that your manuscript would be strengthened if a direct enzymatic activity of PNPLA1 was demonstrated, given that your study is submitted back-to-back with that of Prof Kihara's group, we do not consider this request a prerequisite for further consideration of your manuscript", we do not think it necessary to show the enzymatic activity in our study.

2) Mechanistically, how do the proposed PNPLA1 generated products (OLHFA, EOS, and GlcEOS) regulate keratinocyte terminal differentiation? Can these products compensate for each other for the downstream signaling?

[Answer] In the revised version, we have provided a new data showing that EOS, a PNPLA1-generated primary product, rescued the differentiation defect of PNPLA1^{-/-} keratinocytes. We have not tested the effects of OLHFA and GlcEOS, since they are not commercially available. That means that we need to synthesize them chemically, which will take considerable time. At this stage, we think it sufficient to show the role of EOS, a primary product of PNPLA1, in epidermal differentiation.

Accordingly, the new data has been added to Supplementary Figure 6 and described in Results and Discussion, as follows.

“Cer EOS rescues perturbed differentiation of *Pnpla1*^{-/-} keratinocytes. To further investigate the function of PNPLA1 in keratinocyte differentiation, gene expression in primary keratinocytes prepared from *Pnpla1*^{-/-} and control mice was analyzed in culture. Consistent with the *in vivo* data (Fig. 2a–c), expression of the terminal differentiation marker *Flg* was lower in differentiated *Pnpla1*^{-/-} keratinocytes than in replicate control cells, while that of *Ppard* or *Hbegf* was significantly elevated in differentiated *Pnpla1*^{-/-} keratinocytes (Supplementary Fig. 6). Supplementation of the differentiation medium with EOS(C30:0) partially reversed the altered expression of *Flg*, *Ppard*, and *Hbegf* in *Pnpla1*^{-/-} keratinocytes (Supplementary Fig. 6). These results suggest that the PNPLA1 product

EOS or its derivative(s) regulates terminal keratinocyte differentiation by modulating PPAR δ expression.” (page 13, last paragraph)

“It is tempting to speculate that the increased levels of EOS at the SG/SC border could provide a critical signal for keratinocyte maturation to corneocytes. In our study using cultured *Pnpla1*^{-/-} keratinocytes, the supplementation with EOS reversed the decreased expression of filaggrin and increased expression of HB-EGF towards normal levels. In support of this observation, expression levels of late keratinocyte differentiation markers are reduced in several other knockout mouse lines deficient in the pathway leading to EOS synthesis (e.g. *Elovl4*^{-/-}, *Cers3*^{-/-} and *Abhd5*^{-/-}).^{22,54,55} Moreover, application of synthetic pseudo-acylceramide or GlcEOS recovers diminished barrier function *in vivo* and promotes maturation of cultured keratinocytes by facilitating cornification and CE formation^{56,57}. The existence of such cross-talk between acylceramide metabolism and transcriptional control of keratinocyte differentiation would be advantageous for the coordinated formation of corneocytes and intercellular lamellar membranes that comprise the SC with competent permeability barrier function, although full understanding of the underlying mechanism needs further elucidation.” (page 17, last paragraph)

3) What is the mechanism controlling the observed increase in mRNA levels of lipid metabolism related genes associated with ARCI?

[Answer] In the revised version, we have provided a new data showing that the increased expression of PPAR δ was linked, at least in part, to the upregulation of ARCI-associated lipid-related genes in *Pnpla1*^{-/-} skin. This point has been added to Figure 2c and described in the Results and Discussion, as follows.

“Moreover, expression of PPAR δ and its target genes such as *Fabp5* and *Sprr1b*³⁵ was markedly increased in *Pnpla1*^{-/-} skin relative to WT skin (Fig. 2c and Supplementary Fig. 3a), indicating that PPAR δ is hyperactivated in the absence of PNPLA1. Activation of EGF receptor controls keratinocyte cell fate by downregulating the expression of differentiation-related genes including filaggrin and loricrin⁴². Indeed, heparin-binding EGF-like growth factor (HB-EGF), a potent autocrine growth factor for keratinocytes and putative target gene of PPAR δ ⁴³, was profoundly upregulated in *Pnpla1*^{-/-} epidermis (Fig. 2c and Supplementary Fig. 3a), suggesting that EGF receptor signaling contributes, at least in part, to epidermal hyperplasia and altered keratinocyte differentiation in mutant mice.” (page 9, last paragraph)

“The increased expression of PPAR δ in *Pnpla1*^{-/-} epidermis could explain, at least in part, the induction of a panel of lipid metabolism-related genes associated with ARCI. Indeed, PPAR δ contributes to up-regulation of ABCA12 and GBA in keratinocytes, and PPAR δ deficiency decreases lipid metabolism required for lamellar membrane formation and thereby skin barrier function⁵¹⁻⁵³.” (page 17, the latter half of 2nd paragraph)

Other points:

4) In Figure 1, localization of PNPLA1 in newborn skin was shown, does the localization change in adult skin tissue?

[Answer] In response to this comment, we have stained PNPLA1 in adult mouse skin (Supplementary Figure 1c). We have added this point to the Results, as follows.

“In adult mouse skin, the localization of PNPLA1 in the epidermis was essentially the same as that in newborn skin (Supplementary Fig. 1c).” (page 6, lines 9-10)

5) Fig 2F- Why does the PNPLA1 KO have thicker epidermis?

[Answer] In the field of skin biology, it is generally known that the epidermis becomes thicker when the skin barrier is disrupted. This phenomenon has been considered to be a compensatory adaptation to minimize the barrier loss.

Mechanistically, we have shown that PNPLA1 deficiency resulted in upregulation of the EGF family members and S100 proteins (please see the microarray data in Supplementary Figure 3a), which are known to promote keratinocyte growth and epidermal hyperplasia. We have added these points to the Results, as follows.

“.....epidermal hyperplasia (Fig. 1f, right), which is considered to be an adaptive response to barrier disruption.” (page 7, lines 5–6 from the bottom)

“Moreover, expression of PPAR δ and its potential target genes such as *Fabp5* and *Sprr1b*³⁵ were markedly increased in *Pnpla1*^{-/-} skin relative to WT skin (Fig. 2c and Supplementary Fig. 3a), indicating that PNPLA1 deficiency leads to hyperactivation of PPAR δ . Activation of EGF receptors has been shown to control keratinocyte proliferation and differentiation with decreased expression of differentiation-related genes including filaggrin and loricrin⁴². Indeed, heparin-binding EGF-like growth factor (HB-EGF), a potent autocrine growth factor for keratinocytes and putative target gene of PPAR δ ⁴³, was robustly upregulated in *Pnpla1*^{-/-} epidermis (Fig. 2c and Supplementary Fig. 3a), suggesting that EGF receptor signaling contributes, at least in part, to epidermal hyperplasia and altered keratinocyte differentiation in mutant mice.” (page 9, last paragraph)

6) Fig 2G- Comment on increased number of lamellar sheets in PNPLA1 KO mutant?

[Answer] We have commented on this in the Results.

“Nile red staining of the *Pnpla1*^{+/+} epidermis showed wavy lipid multilayers characteristic of SC intercellular lipid lamellae, whereas granular-like lipid aggregates were present within increased number of densely packed lamellar sheets in the *Pnpla1*^{-/-} epidermis (Fig. 1g), suggesting that keratinocytes are hyperproliferative and defective in the secretion and/or composition of SC lipids in mutant mice.” (page 7, bottom to page 8, top)

7) Fig 3A- The immune response-related genes are not relevant in this figure.

[Answer] Figure 3A has been moved to Supplementary Figure 3a, as suggested by the reviewer #2. Although this reviewer says that the immune response-related genes are not relevant to this study, we have decided to hold this data in the figure, since the data would be informative for future studies on the role of PNPLA1 in immune responses associated with skin barrier dysfunction in the context of atopic dermatitis and psoriasis. We have added the importance of this point as follows.

“It is likely that the enhanced expression of inflammatory cytokines and chemokines is a secondary effect resulting from impaired barrier function, since similar changes have also been observed in several genetically distinct mouse models with barrier defects³³⁻³⁸ and patients with skin diseases such as ichthyosis, atopic dermatitis, and psoriasis^{39,40}.” (page 9, lines 2-6)

However, if the editor still thinks that this data is unnecessary, we will follow the editor’s decision and omit the data from the Figure and text.

8) Fig S3 -Viral transduction of PNPLA1 in air-lift cultures of PNPLA1 KO cells should be performed to rescue the effects on Lucifer yellow and H&E staining.

[Answer] As suggested by the reviewer #2, we have deleted this Figure.

9) Fig 4A- What is the reason for elevated cholesterol and TG in PNPLA1 KO?

[Answer] Our microarray study has shown the increased expression of biosynthetic enzymes for cholesterol and TG. This point has been added to the Results and Discussion.

“These increases in cholesterol, VLCFAs and several ceramide species resulting from *Pnpla1* deficiency accorded with the elevated expression levels of genes related to lipid metabolism such as *Hmgcr*, *Elovl4*, and *Degs2* (Supplementary Fig. 3a,b), suggesting compensatory adaptation of the *Pnpla1*^{-/-} epidermis to the impaired acylceramide synthesis and barrier formation.” (page 12, lines 4-8)

“Although the increase in TG content in *Pnpla1*^{-/-} epidermis at P0 may be explained by the induction of lipogenic enzymes (Fig. 3a, Supplementary Fig. 3a)” (page 14, lines 11–12 from the bottom)

10) Page 11, 2nd paragraph talking about cholesterol and FFA should be fig S5a not 5a.

[Answer] We apologize this careless mistake. We have corrected it.

11) Fig 5a- Stomach mRNA levels should be included to show the specificity of the gene targeting since stomach has the second highest expression in WT (Fig1a).

[Answer] We have added the data to Figure 4a. Actually, there was a small reduction of *Pnpla1* expression in the stomach of *K14*-driven *Pnpla1*^{-/-} mice (as *K14* is also expressed in the gastric epithelium), but we do not think that the small change in this tissue is problematic to evaluate the role of PNPLA1 in the skin, since global heterozygous *Pnpla1*^{+/-} mice did not display any skin phenotypes. We have added this point to the Results.

“Relatively minor reduction of *Pnpla1* expression was also evident in the stomach, in which the *K14* promoter is active⁴⁶, yet it is unlikely that this reduction could influence the skin phenotype since global heterozygous *Pnpla1*^{+/-} mice showed no abnormality.” (page 12, last sentence to page 13, top)

12) In Figure 4a there is clear increase in TG levels in PNPLA1 KO, but in figure 6a there is a minute increase in TG in PNPLA1 KO. These two figures are not consistent.

[Answer] This was due to the difference in ages (E18.5 and P0). Please see our *preliminary* data attached here. We have amended the text accordingly.

“Although the increase in TG content in *Pnpla1*^{-/-} epidermis at P0 may be explained by the induction of lipogenic enzymes (Fig. 3a, Supplementary Fig. 3a), the distinct impact of *Abhd5* and *Pnpla1* ablations on TG levels at E18.5 lends further support to segregation of PNPLA1 from bulk TG hydrolysis in which ABHD5 participates.” (page 14, lines 11-14)

Reviewer #2 (expert in skin barrier)

Remarks to the Author:

The manuscript by Hirabayashi et al., titled "PNPLA1 has a crucial role in skin barrier function by directing acylceramide biosynthesis" has been reviewed.

The authors demonstrate PLPLA1 deficiency leads to neonatal lethality in mice, strongly correlated with skin barrier defects and change in lipid composition and ultrastructural evidence for altered lipid lamellar membranes. Lipid composition profile revealed decrease acylceramides (AcylCer), acylglucosylceramides (AcylGlcCer), and O-acyl-omega-hydroxy fatty acids, as well as increase in presumed precursor lipid species (omega-hydroxy GlcCer) and omega-hydroxy fatty acids. The authors conclude that PNPLA1 catalyzes the omega-O-esterification step (with linoleic acid) to form critical AcylCer in the epidermis.

The results and presentation are both high quality, and the work is original and highly significant. Authors generated a PNPLA1 knockout mouse that showed neonatal lethality, with severe skin defect, altered omega-O-esterification (reduced O-acyl-ceramides), and abnormal epidermal differentiation. Combined data demonstrate the requirement for PNPLA1 in generation of acylCer in epidermis, one of the last key steps in the formation of these unique and critical lipids. Statistics and references cited appear appropriate, and conclusions justified by data presented.

[Answer] We do thank you for giving us very positive comments and suggestions. As indicated in the response that follows, we have taken all these comments and suggestions into account in the revised version of our paper.

General Comments:

1) Manuscript appears to have dual focus: 1) effect of decreased PNPLA1 on epidermal function and lipid composition, revealing key enzymatic activity; and 2) characterization of the PNPL1 knockout. Perhaps better if authors could focus manuscript on item #1; this would reduce need for much of the differentiation-related information that appears to distract from key point of manuscript.

[Answer] Thank you very much for your suggestion. In this revision, we have put a main focus on 1) epidermal function and lipid composition. Kihara's group has elegantly demonstrated the key enzymatic activity (back-to-back paper), so we do not think it necessary to state it here (as pointed out by the editor). With regard to 2) characterization of the PNPL1 knockout in terms of epidermal differentiation, yes we agree that it is not a major point of this study and we have weakened it in the revised version. Nevertheless, in response to the reviewer #1, we have added some new data that could explain the differentiation defect of *Pnpla1*^{-/-} epidermis to Supplementary Figure 6. Please see our answers to the comments 2) and 3) by reviewer #1.

2) Altered epidermal differentiation can itself lead to altered skin barrier function. The authors should comment that the neonatal lethality is likely dependent upon both factors (altered lipid content and abnormal differentiation) as potentially independent events.

[Answer] We have added this point to the Results section.

“Abnormal terminal differentiation of keratinocytes has also been observed in several mouse lines with targeted disruption of genes implicated in epidermal ceramide metabolism^{22,35}. Therefore, the neonatal lethality of *Pnpla1*^{-/-} mice due to skin barrier defect is likely dependent upon both altered lipid composition and impaired terminal differentiation of keratinocytes.” (page 9, 2nd paragraph)

3) Authors did not directly address mechanism by which lack of PLPLA1 alters the differentiation pattern and expression of differentiation-related genes; as such, the question of possible additional activities of PLPLA1, or action(s) of the products of its now apparent enzymatic activity in epidermis, or that of an accumulating precursor, contributing to the skin abnormalities remain unresolved. Inclusion of appropriate comments on this issue appears necessary.

[Answer] We have provided new data and Discussion regarding the mechanism by which the lack of PLPLA1 alters the differentiation pattern and expression of differentiation-related genes. Please see our answers to the comments 2) and 3) by reviewer #1.

4) Results with ABCA12 (-/-) appears to add a layer of complexity to this work that is not critical for the above-noted focus (on PNPL1), especially for the more general audience; suggest removal of these data and discussion to focus manuscript.

[Answer] We have removed the data and discussion on *Abhd12*^{-/-} mice.

Minor comments:

Figure 1: these data are largely represented in Figure 2a (+/+); i.e., showing localization. As such, Fig 1 could be moved to supplemental data.

[Answer] We have moved Figure 1 to Supplemental Figure 1.

Figure 3a: Move to supplemental data; Move Supplemental Fig 2b to Fig 3b (add on).

[Answer] We have moved these Figures as suggested by this reviewer.

Supplementary Figure 3: Unclear why reconstituted (-/-) shows no evidence of hyperplasia. Suggest remove this figure.

[Answer] We have deleted it.

Figure 4f: Remove or move to supplementary data, as does not appear to add significantly.

[Answer] We have moved it to Supplementary Figure 4d.

REVIEWERS' COMMENTS:

Reviewer #1 (Remarks to the Author):

Most of the comments/concerns have been addressed, but one major and 2 minor points need additional attention.

Results obtained by addition of EOS to culture system suggests that EOS at high concentrations can biologically correct the defects in the expression of differentiation related genes. However, this experiment does not provide any insight as to mechanistically, how EOS modulates skin differentiation. This point should be discussed more elaborately. In addition, in order to keep the manuscript coherent, it is advised that these results be mentioned as a part of main figures.

Other points:

Data for immune response-related genes can be kept as a supplementary figure with the provided explanation in the text.

Stomach mRNA expression of Pnpla1 is shown in the gene targeted mice. However, the change appears to be about 50% compared to the WT, not "a small reduction" as described by the authors. Please correct the text with proper description of the data.

Reviewer #2 (Remarks to the Author):

The Authors have addressed each of this Reviewer's concerns, and the manuscript is significantly improved. This Reviewer has no additional comments for the Authors.

Responses to Reviewers

Reviewer #1

Most of the comments/concerns have been addressed, but one major and 2 minor points need additional attention.

Results obtained by addition of EOS to culture system suggests that EOS at high concentrations can biologically correct the defects in the expression of differentiation related genes. However, this experiment does not provide any insight as to mechanistically, how EOS modulates skin differentiation. This point should be discussed more elaborately. In addition, in order to keep the manuscript coherent, it is advised that these results be mentioned as a part of main figures.

[Answer] We have discussed this point in the Discussion, as follows (a paragraph on pages 17-18).

“It is tempting to speculate that the increased extracellular levels of EOS or its derivatives at the SG/SC border could provide a critical signal for keratinocyte maturation to corneocytes. In our study using cultured *Pnpl1*^{-/-} keratinocytes, the supplementation with EOS reversed the decreased expression of filaggrin and increased expression of HB-EGF towards normal levels. In support of this observation, application of synthetic pseudo-acylceramide or GlcEOS recovers diminished barrier function *in vivo* and promotes maturation of cultured keratinocytes by facilitating cornification and CE formation^{55,56}. Moreover, markers for keratinocyte proliferation and differentiation are dysregulated in several other knockout mouse lines deficient in the pathway leading to EOS synthesis, processing or transport (e.g. *Elovl4*^{-/-}, *Cers3*^{-/-}, *Abhd5*^{-/-}, *Abca12*^{-/-}, and epidermal-specific *Ugcg*^{-/-})^{22,35,41,57,58}. In contrast, keratinocyte differentiation is not profoundly affected in *Alox12b*^{-/-} mice⁵⁹, where protein-bound lipids, but not free ceramides including acylceramide, are decreased, consistent with the view that the LOX-catalyzed oxidation of the linoleate residue in acylceramide is required for subsequent ester hydrolysis and covalent binding of the resultant free ω-OH Cer to the CE⁵⁰. These differences could be explained if differentiated keratinocytes have the ability to sense an extracellular pool of EOS or its derivatives through a putative receptor, transporter or other mechanisms. Nonetheless, the existence of such cross-talk between acylceramide metabolism and transcriptional control of keratinocyte differentiation would be advantageous for the coordinated formation of corneocytes and intercellular lamellar membranes that comprise the SC with competent permeability barrier function, although full understanding of the underlying mechanism needs further elucidation.”

We have moved the corresponding data from the Supplementary file to Figure 5 (as a main figure). Accordingly, previous Figure 5 is shifted to Figure 6.

Other points:

Data for immune response-related genes can be kept as a supplementary figure with the provided explanation in the text.

[Answer] Yes, we show the data for immune response-related genes as a supplementary figure with the provided explanation in the text. Thank you.

Stomach mRNA expression of Pnpla1 is shown in the gene targeted mice. However, the change appears to be about 50% compared to the WT, not “a small reduction” as described by the authors. Please correct the text with proper description of the data.

[Answer] Agree. We have changed it to “about half reduction”.

Reviewer #2

The Authors have addressed each of this Reviewer's concerns, and the manuscript is significantly improved. This Reviewer has no additional comments for the Authors.

[Answer] Thank you very much!